# Integrative analysis of epigenetic subtypes in acute myeloid Leukemia: A multi-center study combining machine learning for prognostic and therapeutic insights

Jincan Li, Shengyue Wang ᴼᴿᶜᴵᴰ *

Shanghai Institute of Hematology, State Key Laboratory of Medical Genomics, National Research Centre for Translational Medicine at Shanghai, Ruijin Hospital affiliated to Shanghai Jiao Tong University School of Medicine, Shanghai, China

* wsy12115@rjh.com.cn

## Abstract

### Background

Acute Myeloid Leukemia (AML) exhibits significant heterogeneity in clinical outcomes, yet current prognostic stratification systems based on genetic alterations alone cannot fully capture this complexity. This study aimed to develop an integrated epigenetic-based classification system and evaluate its prognostic value.

### Methods

We performed multi-omics analysis on five independent cohorts totaling 1,103 AML patients. The Cancer Genome Atlas-Acute Myeloid Leukemia (TCGA-LAML) cohort (n = 83) provided comprehensive multi-omics data including DNA methylation profiles (Illumina 450K platform), RNA sequencing (mRNA, lncRNA, and miRNA), and somatic mutation profiles. The BEAT (n = 649), TARGET (n = 156), GSE12417 (n = 79), and GSE37642 (n = 136) cohorts contributed transcriptome data. Molecular subtypes were identified using empirical Bayes-based clustering on the TCGA cohort. LSC17 scores were calculated using a validated 17-gene expression signature. A random survival forest model was developed integrating molecular features with LSC17 scores, validated across all cohorts. Immune microenvironment analysis employed multiple deconvolution methods (ESTIMATE, CIBERSORT, xCell) and pathway analysis (GSVA, GSEA). Drug sensitivity was predicted using the pRRophetic algorithm with GDSC database reference.

### Results

Multi-omics integration revealed two molecularly distinct AML subtypes with significant survival differences (CS2 vs CS1, P < 0.001). The random survival forest model, incorporating 20 key epigenetic features (including CPNE8, CD109, and CHRDL1)

**Data availability statement:** The data used in this study are available from the following public repositories: 1. TCGA-LAML dataset is available from the Genomic Data Commons portal (GDC) Data Portal (https://portal.gdc.cancer.gov/) by searching for Project ID 'TCGA-LAML'. This corresponds to dbGaP Study Accession phs000178. 2. BEATAML dataset is available from the GDC Data Portal (https://portal.gdc.cancer.gov/) by searching for Project ID 'BEATAML1.0-COHORT'. This corresponds to dbGaP Study Accession phs001657. 3. TARGET-AML dataset is available from the GDC Data Portal (https://portal.gdc.cancer.gov/) by searching for Project ID 'TARGET-AML'. This corresponds to dbGaP Study Accession phs000218. 4. Gene expression datasets are available from the Gene Expression Omnibus (GEO) database (https://www.ncbi.nlm.nih.gov/geo/, accession numbers: GSE12417 and GSE37642) 5. Epigenetics-related gene information was obtained from the EpiFactors database (https://epifactors.autosome.org/) The data from these public repositories can be downloaded free of charge for research purposes.

**Funding:** The author(s) received no specific funding for this work.

**Competing interests:** The authors have declared that no competing interests exist.

and LSC17 scores, achieved superior prognostic accuracy (C-index: 0.72–0.78) across validation cohorts. Both epigenetic risk score (HR = 2.45, 95%CI: 1.86–3.24) and LSC17 score (HR = 1.89, 95%CI: 1.42–2.51) maintained independent prognostic value in multivariate analysis. Integration of both scores in a nomogram improved 1-, 3-, and 5-year survival predictions (C-index: 0.81). High-risk patients exhibited distinct immune profiles with elevated M2 macrophages (1.8-fold) and Tregs (2.3-fold), while low-risk patients showed enhanced NK cell activity (2.1-fold). Drug sensitivity analysis identified differential responses to epigenetic regulators (LAQ824, P = 0.000139; MS-275, P = 0.00104) and proteasome inhibitors (Bortezomib, P = 0.00747; MG-132, P = 0.0106) between risk groups.

## Conclusions

This integrated classification system combining epigenetic features and stem cell signatures provides new insights into AML heterogeneity and therapeutic targeting. The complementary nature of epigenetic and stem cell-related prognostic factors suggests potential for improved risk stratification in clinical practice. Future prospective validation studies are warranted to confirm these findings.

## Introduction

Acute Myeloid Leukemia (AML), as a hematological malignancy [1], is characterized by disordered proliferation of myeloid precursor cells. It represents one of the most common types of leukemia in adults, with an incidence rate showing a significant upward trend alongside population aging [2]. Despite continuous improvements in conventional therapeutic approaches, including intensive chemotherapy, targeted therapy, and hematopoietic stem cell transplantation, patient outcomes remain unsatisfactory due to the high heterogeneity and complex molecular mechanisms of the disease, particularly among elderly and high-risk populations [3]. Clinical data indicate that over 50% of patients who achieve remission eventually relapse [4], highlighting the urgency of identifying novel therapeutic strategies. Currently, AML diagnosis and prognostic assessment primarily rely on cytogenetic and molecular biological testing [1,5]; however, these traditional methods often fail to fully reflect the disease's complexity and heterogeneity, limiting the accuracy of treatment outcome predictions [6].

Recent advances in epigenetics research have provided new perspectives in understanding AML pathogenesis [7]. Epigenetic regulatory mechanisms, including DNA methylation, histone modifications, and non-coding RNAs, play crucial roles in AML development, progression, and prognosis [8]. These epigenetic alterations not only affect leukemic cell proliferation, differentiation, and survival but are also closely associated with clinical phenotypes and outcomes [9,10]. Studies have shown that aberrant DNA methylation patterns may lead to silencing of key tumor suppressor genes or activation of oncogenes, while changes in histone modifications can affect chromatin structure and gene expression regulation. Furthermore, long non-coding RNAs and microRNAs play important roles in leukemic stem cell maintenance and

disease progression [11,12]. However, the potential of epigenetics in developing individualized prognostic models remains underexploited, particularly in the context of multi-omics data integration.

With the advancement of high-throughput sequencing technologies, multi-omics research approaches have provided unprecedented opportunities for understanding AML in depth [13]. The integration of genomics, transcriptomics, and epigenomics data enables a more comprehensive mapping of AML's molecular characteristics. These multidimensional data not only help understand the molecular mechanisms of disease development but also provide important clues for developing new therapeutic targets. Particularly in resistance mechanism studies [14], analyzing epigenetic changes and gene expression profile alterations before and after treatment can identify key factors affecting treatment response, providing new strategies to overcome drug tolerance. The application of artificial intelligence and machine learning algorithms has made it possible to extract meaningful biological information from massive multi-omics data. By establishing prediction models that integrate multi-omics data, more accurate assessment of patient prognosis and guidance for individualized treatment planning can be achieved. This systems analysis approach based on multidimensional data considers both genetic and epigenetic alterations while integrating clinical features and treatment response information, potentially improving prognostic prediction accuracy and providing crucial support for precision medicine implementation. However, to fully realize the potential of multi-omics research, key issues such as standardization of data integration and analysis methods, and large-scale clinical validation need to be addressed. Establishing multicenter collaborative research networks to collect more patient samples and clinical information will help validate the universality and reliability of research findings, while developing more advanced bioinformatics analysis tools and algorithms remains an important direction for improving multi-omics data utilization efficiency.

## Materials and methods

### Data source

This study integrated five independent AML cohort datasets. The Cancer Genome Atlas-Acute Myeloid Leukemia (TCGA-LAML) dataset (https://portal.gdc.cancer.gov/, accession number: phs000178) provided the most comprehensive multi-omics data, including expression profiles of mRNA, miRNA, long non-coding RNA (lncRNA), DNA methylation profiles (based on Illumina 450K platform), and somatic mutation information for 83 patients with complete data. Additionally, we integrated data from 649 patients from the BEAT database (https://portal.gdc.cancer.gov/, accession number: phs001657) and 156 AML patients from the Therapeutically Applicable Research to Generate Effective Treatments (TARGET) Project (https://portal.gdc.cancer.gov/, accession number: phs000218) [15]. We also utilized two public gene expression datasets from the Gene Expression Omnibus (GEO) database (https://www.ncbi.nlm.nih.gov/geo/): GSE12417 (79 patients) and GSE37642 (136 patients). All datasets underwent rigorous screening, retaining only samples with complete Overall Survival (OS) follow-up information, resulting in a total sample size of 1,103 cases (Table 1Table 1). This multi-cohort integration design enhanced statistical power and improved the reliability and universality of research findings. Epigenetics-related genes were sourced from the EpiFactors database (https://epifactors.autosome.org/) [16].

### Ethics approval and consent to participate

This study utilized publicly available, de-identified datasets from multiple repositories (TCGA-LAML, BEAT database, TARGET Project, and GEO datasets GSE12417 and GSE37642). All original data collection was conducted with appropriate patient consent and ethical approval by the respective institutions that generated these datasets. As our study involved only secondary analysis of de-identified, publicly available data, additional ethics approval was not required for our specific analyses according to institutional policies. All data usage complied with the data access policies and terms of service for each repository.

### TCGA-LAML data preprocessing and integration

Multi-omics data from the TCGA-LAML dataset underwent systematic preprocessing and integration. Initially, protein-coding gene expression data were processed by normalizing TPM values and averaging multiple transcripts for the same

**Table 1. Overview of datasets, sample sizes, Scources, and molecular features used in this study.**

| Dataset | Sample Size | Scource | Molecular Features |
|---------|-------------|---------|--------------------|
| TCGA-LAML | *83* | Blood | mRNA expression, miRNA expression, lncRNA expression, DNA methylation, Somatic mutations |
| BEAT | 649 | Blood | mRNA expression |
| TARGET | 156 | Blood | mRNA expression |
| GSE12417 | 79 | Blood | mRNA expression |
| GSE37642 | 136 | Blood | mRNA expression |

gene Symbol. Subsequently, long non-coding RNA (lncRNA) and microRNA (miRNA) expression data underwent identical standardization processing, with all expression data transformed by log2(TPM + 1) to achieve approximate normal distribution. DNA methylation data retained beta values from the Illumina 450K platform as methylation level quantification indicators. Somatic mutation data were processed through the maftools package and converted into binary matrices [17]. For clinical data, we extracted key information including patient survival status (OS), survival time, gender, and age, retaining only samples with positive survival times. To ensure data quality, we specifically developed sample screening functions, retaining only primary hematological tumor samples (sample numbers with "03" in positions 14–15), and ensured sample consistency across all omics data through sample ID matching. Finally, processed data were stored in standardized formats for subsequent analysis.

## Multi-dataset expression profile integration and batch effect correction

To integrate multiple transcriptome datasets, we developed a standardized data processing pipeline. Initial preprocessing through the process_data function included calculating arithmetic means for duplicate gene expressions and standardizing sample naming formats. During data integration, we retained only genes common to all datasets and samples with complete survival information (OS > 0). The ComBat algorithm was employed to correct batch effects between different datasets [18], utilizing an empirical Bayesian framework to effectively remove non-biological variation. Data quality control was assessed through Principal Component Analysis (PCA), with corrected expression profiles and clinical information stored in standardized formats.

## Molecular subtyping analysis based on multi-omics data

This study employed the MOVICS package for multi-omics integrated molecular subtyping [19]. Initially, epigenetics-related genes were screened: mRNA underwent survival correlation screening using Cox proportional hazards models (p < 0.05); lncRNA and methylation data first selected the top 1500 features with highest variation through MAD (Median Absolute Deviation) followed by Cox screening (p < 0.05); miRNA retained the top 50% features by MAD value before Cox screening; mutation data retained genes with frequencies exceeding 5%. Clustering analysis utilized empirical Bayesian-based integrated clustering methods, with Gaussian distribution models for mRNA, lncRNA, miRNA, and methylation data, and binomial distribution models for mutation data. Optimal cluster numbers were determined through consensus matrix cumulative distribution function (CDF) and Delta area. For enhanced visualization, methylation β values were converted to M values, and all data (except mutation data) underwent centralization and standardization. Sample clustering stability was evaluated through Silhouette values.

## Molecular subtype characterization and validation

This study analyzed molecular subtype biological characteristics from multiple perspectives. First, the ssGSEA algorithm from the GSVA package was used to score immune-related pathways [20], including immune cell development functions, immune signaling pathways, and immune effector functions. The RTN package was employed to construct transcription

factor-centered regulatory networks [21], focusing on mucin and chromatin remodeling-related genes, with 1000 permutation tests to ensure network reliability. Detailed characterization of the tumor immune microenvironment combined MeTIL scores [22], ESTIMATE algorithm for stromal and immune infiltration levels [23], and CIBERSORT for immune cell component quantification [24]. To validate subtype stability, NTP and PAM algorithms were used for cross-validation, selecting 100 feature genes per subtype as classification templates. Subtyping results were evaluated through Kappa consistency tests between different methods.

## Machine learning model construction and validation

This study employed various machine learning methods to construct prognostic prediction models. The BEAT dataset, with the largest sample size, served as the training set, while TCGA-LAML, GSE37642, GSE12417, and TARGET datasets served as independent validation sets. All datasets underwent standardization processing, retaining only features common to all datasets. In model construction, we evaluated ten basic models and their combinations, including Random Survival Forest (RSF) [25], Elastic Net [26], CoxBoost [27], and Gradient Boosting Machine (GBM) [28]. Each model underwent 10-fold cross-validation for parameter optimization. The RSF model was set with 1,000 trees and a minimum node size of 5; Elastic Net's α parameter underwent grid search between 0.1–0.9 with 0.1 steps; GBM models employed Cox proportional hazard distribution. The best-performing model underwent detailed analysis, with key features selected through variable importance scores. Optimal cutpoint methods divided patients into high and low-risk groups, with model predictive performance evaluated through Kaplan-Meier survival analysis and time-dependent ROC curves [29].

## Molecular pathway enrichment analysis and biological function annotation

This study adopted a multi-level enrichment analysis strategy to explore the biological mechanisms of the prognostic model. Initially, the limma package was used for differential expression analysis between high and low-risk groups, with genes sorted by logFC values for subsequent GSEA analysis. GSEA enrichment analysis was performed using Molecular Signatures Database (MSigDB)'s hallmark gene sets (v2023.1) [30], with significance level set at $p < 0.05$.

To obtain a more comprehensive pathway activity landscape, we employed the GSVA algorithm to score hallmark gene sets and compared pathway activity differences between high and low-risk groups using the limma package. Pathway activity differences were assessed through t-tests, with pathway correlations visualized through the corrplot package. Additionally, survival analysis was performed for each pathway, dividing samples into high and low expression groups based on median values, with pathway activity and prognosis associations evaluated through log-rank tests ($p < 0.05$). Finally, significant pathways underwent Cox hazard ratio analysis, with prognostic values displayed through forest plots. All analyses were completed using R 4.1.0, with graphics visualization primarily implemented through ggplot2, survminer, and other packages.

## Immune microenvironment characterization

To further explore immune microenvironment characteristics from multiple dimensions, we employed the ESTIMATE algorithm to evaluate sample stromal scores, immune scores, and overall scores, comparing differences between high and low-risk groups. Multiple immune cell infiltration assessment methods (CIBERSORT, EPIC [31], MCPcounter [32], xCell [33], TIMER [34], and quanTIseq [35]) were used for immune cell composition quantification. For immune-related pathways, we integrated immune-related gene sets from the KEGG database, calculating pathway activity scores for each sample using the ssGSEA algorithm. Immune pathway activity heatmaps were generated using the pheatmap package, with differences between high and low-risk groups evaluated through Wilcoxon rank-sum tests (marked as * for $p < 0.05$, ** for $p < 0.01$, *** for $p < 0.001$). Furthermore, to explore associations between risk scores and immune characteristics, we conducted Spearman correlation analysis between significantly different immune features and risk scores. Additionally, correlations between feature genes and immune cell infiltration were analyzed and visualized using the corrplot package.

These analyses were performed not only in the TCGA-LAML database but also validated in other databases to ensure result reliability.

## Somatic mutation characteristic analysis

This study conducted in-depth analysis of AML patients' somatic mutation characteristics. First, tumor mutation burden (TMB) was calculated using the maftools package, with differences between high and low-risk groups compared through density plots and box plots (Wilcoxon test). Mutation landscape analysis was performed for high and low-risk groups separately, with the oncoplot function displaying the top 20 most frequently mutated genes. Additionally, we analyzed mutation gene commonality and exclusivity within each group (somaticInteractions function), with significance levels set at 0.05 and 0.1.

## Drug sensitivity and treatment strategy analysis

This study used the pRRophetic package to predict AML patients' sensitivity to different drugs [36]. Initially, differential expression analysis and GSEA enrichment analysis identified differential pathways between high and low-risk groups, providing theoretical foundations for drug targeting. GSEA analysis employed MSigDB's hallmark gene sets with significance level set at $p < 0.05$, visualizing enrichment results and core genes through the gseaNb function. Drug sensitivity predictions were based on cell line expression profiles and IC50 data from the Genomics of Drug Sensitivity in Cancer (GDSC) database (https://www.cancerrxgene.org/) [37]. The pRRopheticPredict function was used to predict sensitivity for all available drugs in the cgp2016 dataset (https://rdrr.io/github/xlucpu/MOVICS/man/cgp2016ExprRma.html). For each drug, predicted IC50 value differences between high and low-risk groups were compared through Wilcoxon rank-sum tests. For significantly different drugs ($p < 0.05$), box plots were generated using the ggplot2 package to display differences. Final results were validated across multiple datasets to ensure validity.

## Single-cell transcriptome analysis to validate the biological significance of the model

To explore and validate the biological significance of the RSF risk score model at the single-cell level, this study analyzed single-cell RNA sequencing data from AML patients. The phs000159 dataset [38] was obtained from the ABC portal database (http://abc.sklehabc.com/). This dataset was first generated by Allegra A. Petti et al. in 2019 using the 10X Genomics platform and published in Nature Communications. It contains single-cell sequencing results of 87,538 bone marrow samples from AML patients [39].

After importing the pre-processed matrix to construct a Seurat object, we integrated the pre-calculated UMAP dimensionality reduction coordinates and cell type annotation information to build a complete single-cell analysis framework. To distinguish between malignant and non-malignant cells, the cell malignancy status annotation information was imported and integrated into the analysis process. Particularly, this study focused on the hematopoietic stem cell (HSC) subpopulation, which is of utmost importance in the hematopoietic system. This key cell population was identified through cell-type markers, and malignant HSC and non-malignant HSC were further distinguished.

To evaluate the role of epigenetic regulation in the pathogenesis of AML, this study utilized the epigenetic-related gene set organized above and calculated the epigenetic regulation activity score for each HSC cell using the ssGSEA method. Subsequently, the characteristics of the spatial distribution of these scores in cells were visualized by UMAP, and the differences in epigenetic activities between malignant and non-malignant HSC were compared using the Wilcoxon rank-sum test.

Most importantly, using the same ssGSEA method, this study calculated the LSC17 score and the expression score of high-weight genes in the RSF model for each HSC cell, respectively. LSC17 is a widely validated gene set related to AML stem cells, containing 17 genes closely related to the functions of leukemia stem cells. The RSF model score is based on the 20 genes with the highest weights in the newly developed random forest model of this study. These two sets of scores represent classic and newly developed prognostic markers of AML, respectively. This study analyzed again using the same statistical scheme and quantitatively compared the difference patterns between malignant and non-malignant HSC.

## Results

The research process is shown in Fig 1.

### Identification and characterization of two molecularly distinct AML subtypes through multi-omics integration

Through integrated analysis of multi-omics data from the TCGA-LAML cohort, we revealed two distinct molecular subtypes of acute myeloid leukemia. Systematic evaluation of clustering prediction indices and gap statistics indicated that the optimal number of multi-omics integrated subtypes was 2 (Fig 2A). Using empirical Bayesian integrated clustering methods, we classified 83 samples into two subtypes, with silhouette analysis showing good clustering stability (average silhouette width = 0.56). Subtype 1 (CS1) included 50 samples (silhouette value = 0.51), while subtype 2 (CS2) contained 33 samples (silhouette value = 0.63) (Fig 2B).

Multi-omics integration analysis revealed systematic differences between these two subtypes at transcriptomic, epigenomic, and genomic levels. The hierarchical clustering heatmap clearly demonstrated unique patterns of mRNA, lncRNA, miRNA, DNA methylation, and somatic mutation features (Fig 2C). To validate classification robustness, we employed multiple clustering algorithms for cross-validation, including SNF, CIMLR, and PINSPlus methods (Fig 2D). Consensus matrix analysis further supported the rationality of binary classification (Fig 2E). Notably, survival analysis indicated that CS2 subtype patients exhibited significantly better OS than CS1 subtype (HR = 2.22 (95%CI, 1.69–3.14), P < 0.001) (Fig 2F), highlighting the potential clinical prognostic value of this classification system.

### Biological characteristics and validation of molecular subtypes

Systematic functional annotation revealed significantly different biological characteristics between the two AML molecular subtypes. Through GSVA analysis, we found these two subtypes exhibited significant differences in multiple key signaling pathways, particularly in NOD-like receptor signaling pathway and PPARG network (Fig 3A). Through transcriptional

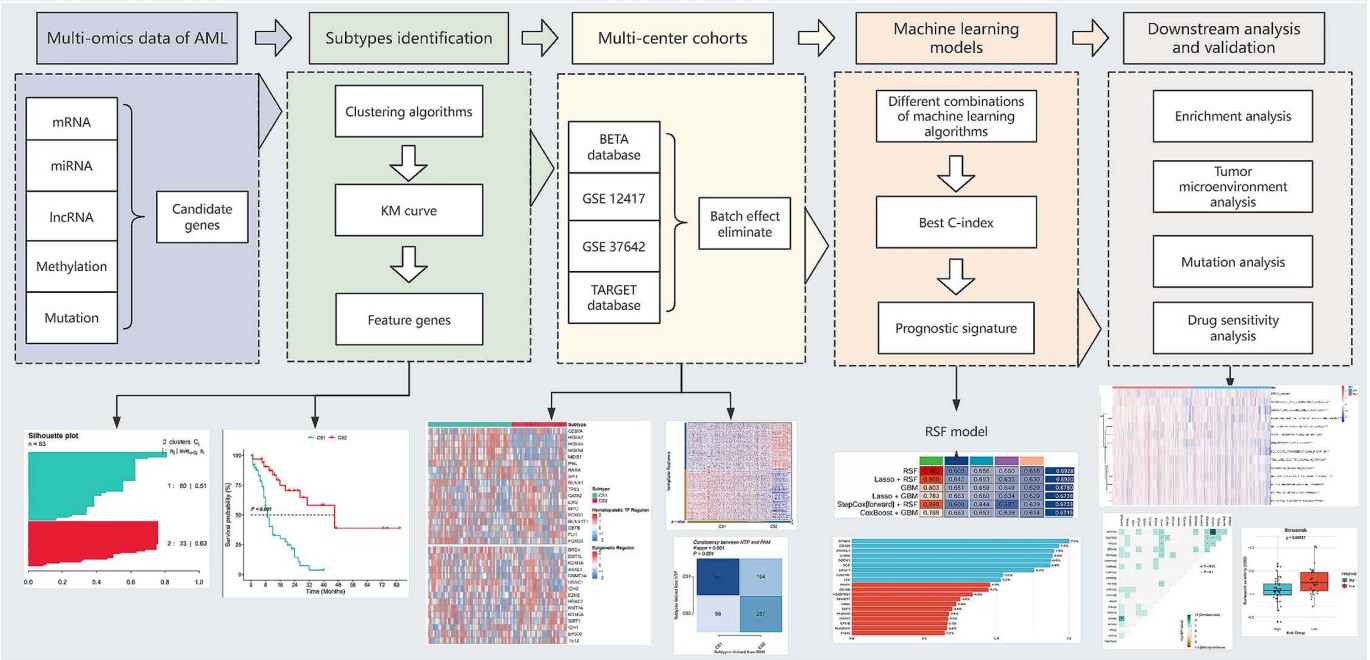

**Fig 1. Study flow.**

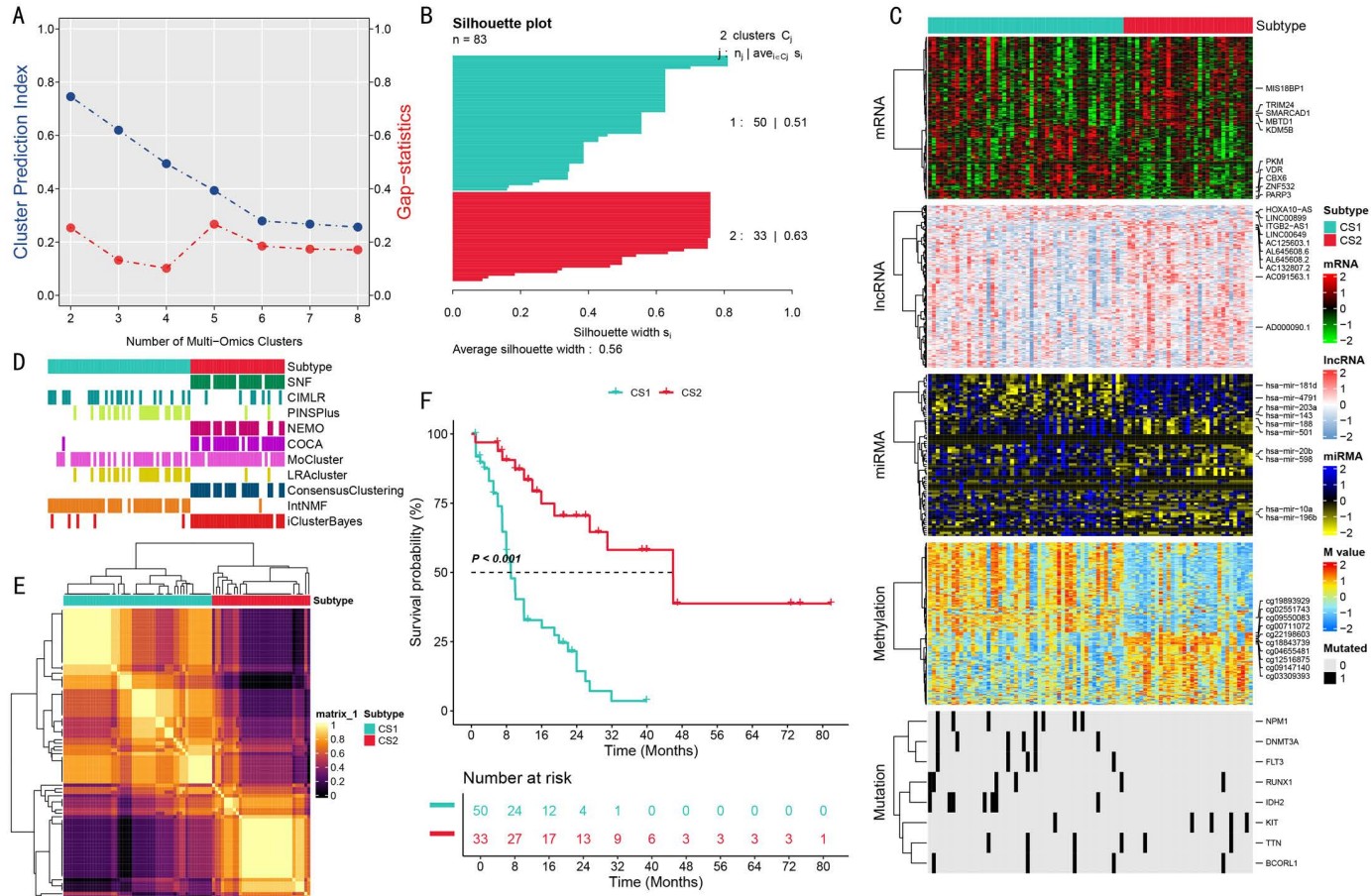

**Fig 2. Multi-omics Molecular Subtyping Analysis of AML. (A)**. Clustering prediction index and gap statistics analysis plot. Blue curve represents clustering prediction index (left y-axis), red curve represents gap statistics (right y-axis), x-axis shows number of multi-omics clusters. **(B)**. Sample silhouette analysis. Shows silhouette coefficient distribution for two subtypes (CS1: n = 50, silhouette value = 0.51; CS2: n = 33, silhouette value = 0.63). **(C)**. Multi-omics integrated clustering heatmap. Displays expression patterns of mRNA, lncRNA, miRNA, DNA methylation, and mutation features from top to bottom. Color scale indicates normalized expression levels. **(D)**. Comparative analysis results of multiple clustering algorithms. Shows classification results from different clustering methods including SNF, CIMLR, PINSPlus. **(E)**. Consensus matrix heatmap. Shows clustering consistency between samples, color from dark to light indicates consistency from low to high. **(F)**. Kaplan-Meier survival analysis curves. Green curve represents CS1 subtype, red curve represents CS2 subtype. Numbers below show risk numbers at each time point.

regulatory network analysis, we identified distinct regulatory patterns centered on two major groups of regulators (Fig 3B). The first group consisted of core hematopoietic transcription factors (including CEBPA, HOXA7/9, HOXB4, MEIS1, PML, RARA, SPI1, RUNX1, and TP53) and showed subtype-specific activity patterns. The second group comprised epigenetic modifiers and chromatin remodeling factors (including BRD4, DOT1L, KDM1A, ASXL1, and DNMT3A), which exhibited distinct regulatory patterns across subtypes.

To deeply understand tumor immune microenvironment characteristics, we performed comprehensive immune profiling through multiple approaches (Fig 3C and D). We analyzed the expression of immune checkpoints (including CD96, CD244, HAVCR2, CD86, LILRB1/2) and evaluated immune cell infiltration using CIBERSORT. The analysis revealed distinct immune cell composition patterns across subtypes, particularly in myeloid cells (monocytes, macrophages M0/ M1/M2), T cell subsets (CD4 + , CD8 + , Tregs), and NK cells. Additionally, we calculated ESTIMATE algorithm scores for immune and stromal components, and the MeTIL index, which together provided a comprehensive view of the immune

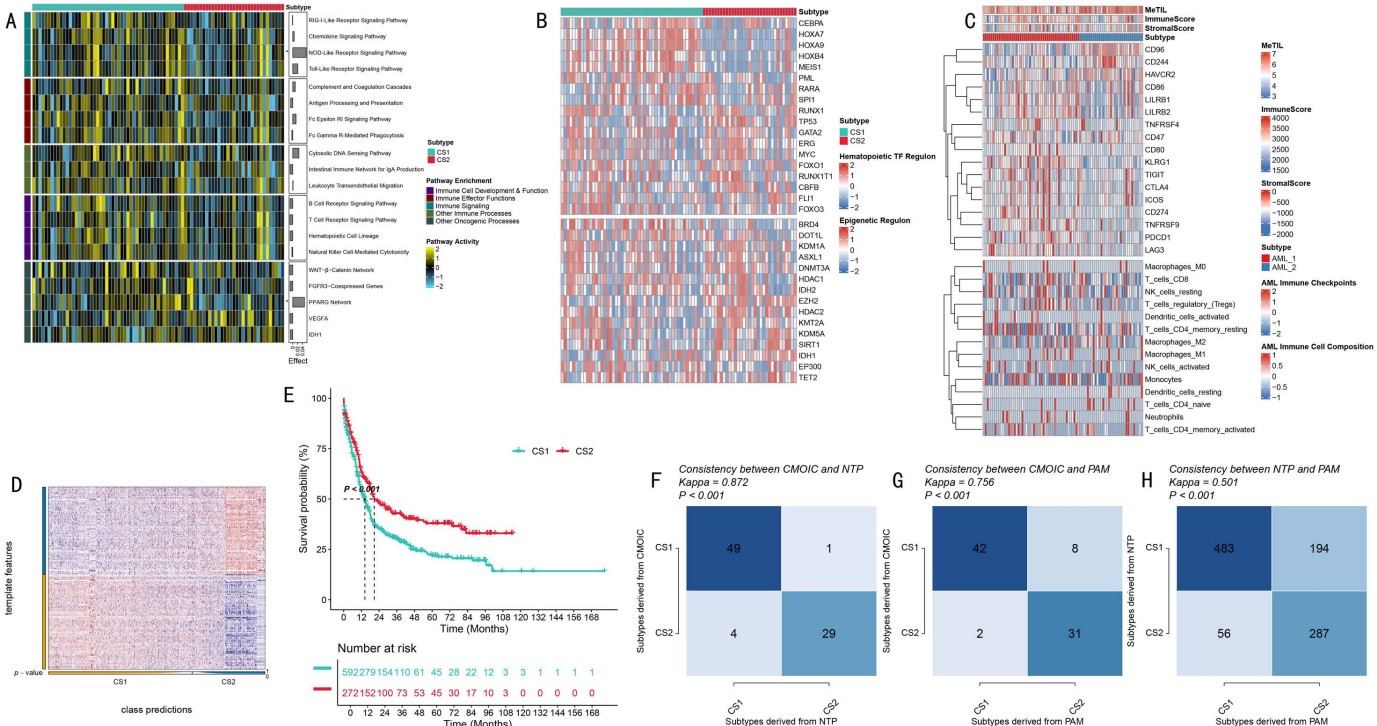

**Fig 3. Biological Characteristics and Validation Analysis of Molecular Subtypes. (A)**. Immune-related pathway activity heatmap. Rows represent different signaling pathways, columns represent samples, color intensity indicates pathway activity level. **(B)**. Transcription regulatory network analysis. Upper panel shows hematopoietic transcription factor regulon activities, lower panel shows epigenetic regulator activities. Colors indicate scaled regulon activity. **(C)**. Tumor immune microenvironment characteristic analysis. From top to bottom: immune checkpoint expression patterns, microenvironment scores, and immune cell composition by CIBERSORT. Color scale represents relative levels. **(D)**. Classification prediction matrix plot. Shows prediction probability distribution of sample classification. **(E)**. Validation cohort survival analysis. Kaplan-Meier curves show survival differences between two subtypes, including risk number table. **(F-H)**. Classification method consistency validation. Shows cross-validation results of CMOIC, NTP, and PAM methods, including Kappa consistency coefficients.

microenvironment. These analyses demonstrated subtype-specific immune characteristics, with notable differences in immune cell infiltration levels and checkpoint expression patterns.

Notably, the CS1 subtype showed higher immune activity and cell infiltration levels. Furthermore, to verify the reliability of subtyping results, we adopted a rigorous cross-validation strategy. In independent validation cohorts, CS2 subtype patients consistently demonstrated significantly better survival benefits than CS1 subtype (HR = 1.61 (95%CI, 1.35–1,92), P < 0.001, Fig 3E). More importantly, through cross-validation using three independent classification methods - CMOIC, NTP, and PAM - we obtained highly consistent classification results (Kappa values of 0.872, 0.756, and 0.501, respectively; Fig 3F-H), strongly supporting the robustness of this molecular subtyping system.

## Development and validation of an RSF-based prognostic model

To construct an accurate prognostic prediction model, we first evaluated the predictive performance of 82 machine learning models (Fig 4A). Through comprehensive comparison of C-indices, the RSF model demonstrated optimal predictive efficiency. Based on variable importance analysis of the RSF model, we identified 20 key predictive features, among which nine genes including CPNE8, CD109, and CHRDL1 showed relative importance significantly higher than 5% (Fig 4B).

In the BETA training set (n = 649), after dividing patients into high and low-risk groups based on optimal cutpoints, Kaplan-Meier analysis revealed significant survival differences between groups (HR = 2.17 (95%CI, 1.65–2.85), P < 0.001, Fig 4C). To rigorously assess the model's generalization ability, we conducted validation in four independent cohorts. In the TCGA-LAML cohort (n = 83), the model similarly showed significant predictive capability (HR = 1.64 (95%CI, 1.03–2.86), Fig 4D). This predictive efficacy was further confirmed in the GSE12417 cohort (n = 79, HR = 3.72 (95%CI, 1.89–7.35), P < 0.001, Fig 4E) and continuously validated in GSE37642 (n = 136, HR = 2.03 (95%CI, 1.33–3.11), P < 0.001) and TARGET cohorts (n = 156, HR = 2.29 (95%CI, 1.42–3.69), P < 0.001) (Fig 4F and G). Notably, analysis of the combined validation cohorts not only confirmed the model's predictive ability but also highlighted result reliability with narrower confidence intervals (HR = 1.60 (95%CI, 1.36–1.88), P < 0.001, Fig 4H).

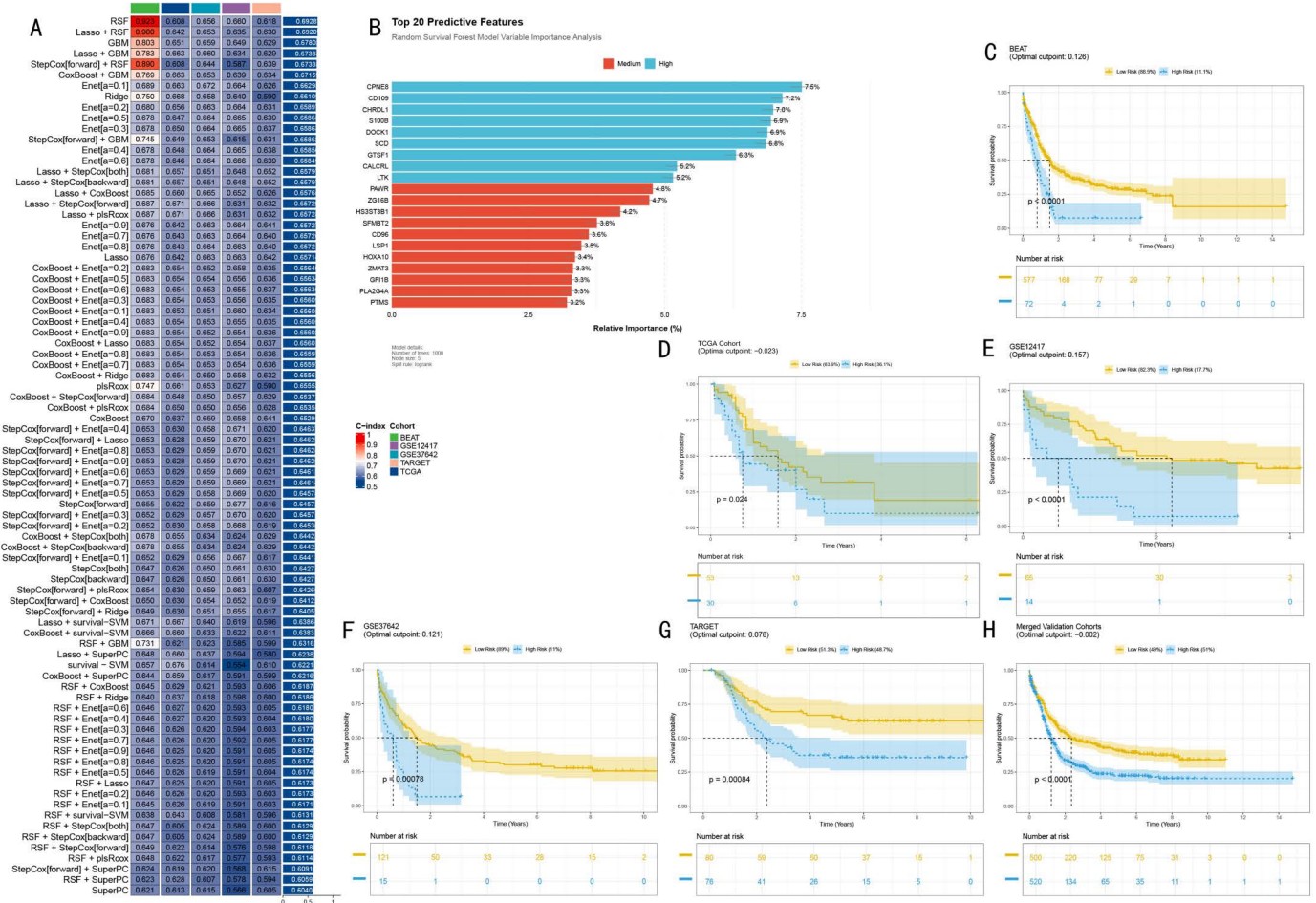

**Fig 4. Construction and Validation of RSF-based AML Prognostic Prediction Model. (A)**. Machine learning model performance heatmap. Shows predictive efficiency of 82 models across different validation sets, color intensity indicates C-index (0-1). Each row represents an algorithm combination, each column represents a validation dataset. **(B)**. Top 20 predictive features of RSF model. Bar chart shows relative importance of key genes, arranged in descending order. Blue bars represent high-importance features (>5%), red bars represent medium-importance features. **(C)**. Kaplan-Meier survival analysis of BETA training set (n = 649). Yellow and blue curves represent high and low-risk groups respectively, shaded areas indicate 95% confidence intervals. Risk table below shows follow-up numbers at each time point. **(D-G)**. Survival analyses of validation sets: TCGA-LAML (n = 83), GSE12417 (n = 79), GSE37642 (n = 136), and TARGET (n = 156), showing model predictive efficiency in independent cohorts. **(H)**. Combined validation cohort survival analysis. Integrates all validation set data to show overall model predictive efficiency, including narrower confidence intervals.

## LSC17 score and molecular pathway analysis reveal biological features of risk groups

To systematically evaluate the prognostic value of risk score and LSC17 score in our prognostic model, we first conducted comprehensive Cox proportional hazards regression analyses. Univariate analysis results (Fig 5A) revealed that both risk score and LSC17 score were significantly associated with prognosis (all P < 0.01), with LSC17 score showing a lower hazard ratio. Multivariate Cox regression analysis further confirmed that both scores possessed independent prognostic predictive value (Fig 5B). Based on their prognostic value, we constructed an integrated nomogram prediction model. Calibration curve analysis assessing the model's predictive accuracy demonstrated excellent calibration for 1-year (red), 3-year (blue), and 5-year (green) survival predictions (Fig 5C). Decision curve analysis (DCA) further confirmed that the integrated nomogram model improved prediction accuracy compared to single prognostic factors (Fig 5D). We established a visualization nomogram incorporating both scores for different years (Fig 5E). Dynamic analysis of time-dependent C-index showed that the nomogram model, after integrating LSC17 score and our Risk score model, consistently outperformed single prognostic factors throughout the follow-up period (Fig 5F). This integrated prognostic prediction tool provides clinicians with an intuitive and accurate individualized prognostic assessment approach.

Based on RSF model risk stratification, we conducted systematic functional enrichment analysis to reveal its molecular biological basis. GSEA showed that the high-risk group was significantly enriched in cancer-related signature pathways, including E2F TARGETS, G2M CHECKPOINT, IL6 JAK STAT3 SIGNALING, INFLAMMATORY RESPONSE, and MYC TARGETS V1 (FDR < 0.05, Fig 6A). GSVA further revealed broader pathway differential characteristics, notably

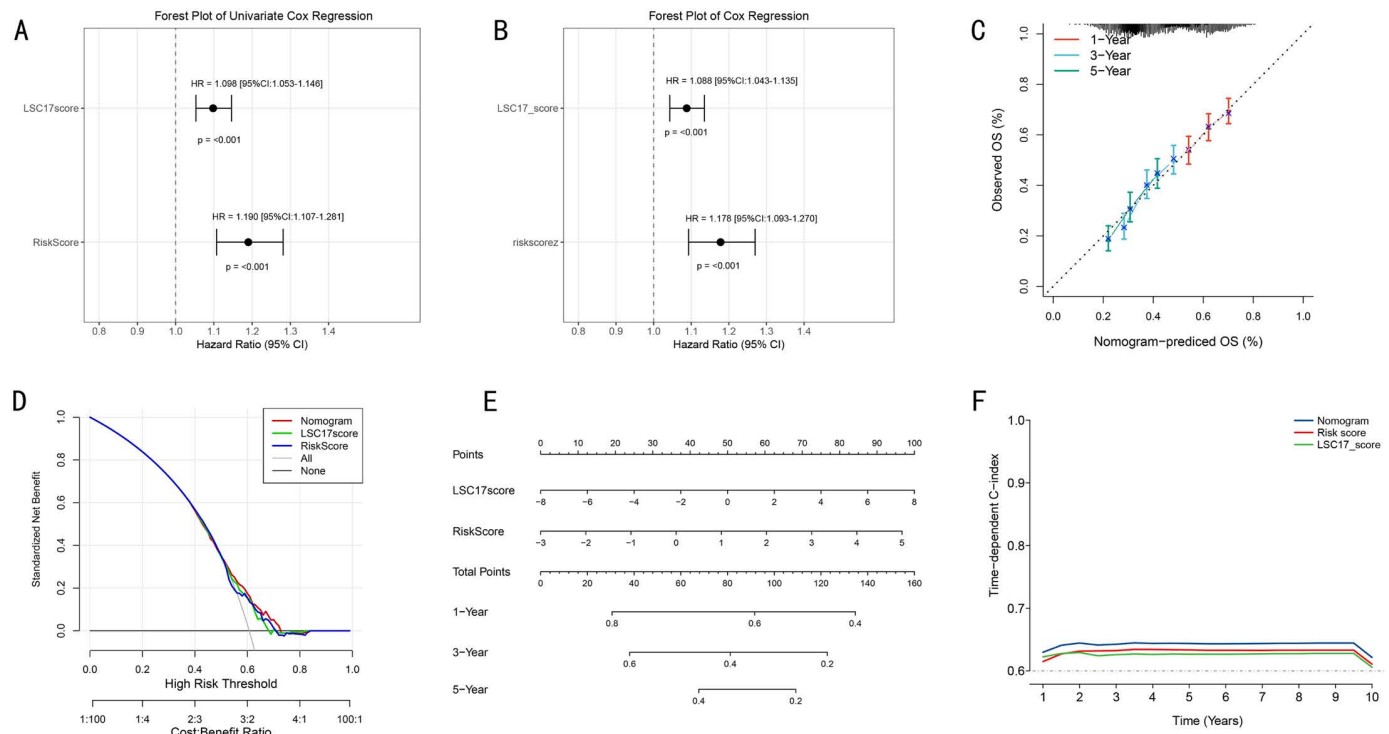

**Fig 5. Construction and Validation of Prognostic Nomogram Based on RSF Model. (A)**. Forest plot of univariate Cox regression analysis showing hazard ratios and 95% confidence intervals for risk score and LSC17 score. **(B)**. Forest plot of multivariate Cox regression analysis confirming independent prognostic factors. **(C)**. Calibration curves of nomogram model for predicting 1-year, 3-year, and 5-year survival probability. **(D)**. Decision curve analysis comparing different prediction strategies. **(E)**. Prognostic nomogram constructed based on risk score and LSC17 score. **(F)**. Time-dependent C-index curves showing dynamic changes of different prediction models.

**Fig 6. Molecular Mechanism Analysis of RSF Model. (A)**. GSEA enrichment analysis plot. Shows key pathways significantly enriched in high-risk group, including enrichment scores and statistical significance. **(B)**. GSVA differential pathway analysis. Waterfall plot shows significantly different biological pathways between high and low-risk groups, sorted by t-value. **(C)**. Risk score and pathway activity correlation heatmap. Blue indicates negative correlation, red indicates positive correlation, intensity shows correlation strength. **(D)** Pathway hazard ratio forest plot. Shows hazard ratios (HR) and 95% confidence intervals for key pathways, sorted by HR value.

the significant activation of functional pathways such as estrogen/androgen response pathways in the high-risk group, warranting further investigation in future studies (Fig 6B). Through correlation analysis between risk scores and pathway activities, we constructed a comprehensive functional regulatory network map (Fig 6C). To assess the clinical significance of these differential pathways, we selected 26 key signaling pathways from the hallmark pathway set that showed statistical significance in Cox proportional hazards model analysis, and further quantified their detrimental effects, with all key pathways demonstrating significant adverse prognostic effects (HR > 1, Fig 6D).

## Comprehensive analysis of tumor microenvironment features

For systematic evaluation of AML tumor microenvironment characteristics, we first conducted multidimensional analysis in the META integrated dataset. ESTIMATE algorithm showed significantly elevated microenvironment scores in the high-risk group, including stromal score (P = 2.1e-07, Fig 7A), immune score (P = 5.8e-09, Fig 7B), and comprehensive score (P = 3.8e-09, Fig 7C). Further immune cell component analysis revealed significant differences between high and low-risk groups in microenvironment cell types (Fig 7D), immune suppression (Fig 7E), immune rejection (Fig 7F), and immune marker expression (Fig 7G). Immune-related pathway analysis identified 14 significantly different signaling pathways (Fig 7H), covering key biological processes including receptor signal transduction (BCR/TCR/FcεRI/Fcγ), pattern recognition receptor signaling (NOD/Toll-like receptor), cell migration (leukocyte transendothelial migration, chemokines), and immune effector functions (antigen processing and presentation, complement cascade, NK cell cytotoxicity).

To validate these findings' reliability, we conducted independent validation in the TCGA-LAML cohort. Analysis results showed highly consistent immune microenvironment characteristics with the integrated dataset, including immune cell component profiles (Fig 8A) and immune suppression and rejection features (Fig 8B-D). Notably, we confirmed differential expression patterns of 6 key immune pathways in the validation set (Fig 8E), including NOD/Toll-like receptor signaling transduction, FC Gamma R-mediated phagocytosis, chemokine signaling, and cytosolic DNA sensing pathways, further supporting these pathways' important roles in AML prognostic stratification.

## Genomic landscape and tumor heterogeneity analysis

Through integration of META and TCGA-AML datasets, we initially identified 57 common immune cell features, providing a reliable feature set for subsequent analysis (Fig 9A). Correlation analysis between these features and the top 20 predictive genes in the prognostic model revealed a complex regulatory network: LSP1, as a key cytoskeletal protein, showed significant positive correlations with most immune cell features, while immune checkpoint molecule CD96 and tyrosine kinase receptor LTK exhibited widespread negative correlation patterns (Fig 9B). Quantitative correlation analysis further refined these associative characteristics: dendritic cells (Dendritic_cells_quantiseq), NK cells (NK_cells_Bindea_et_al), stromal cells (ecm_myCAF), and tumor microenvironment scores (TMEsCoreB CIR) showed significant positive correlations with prognosis. In contrast, multiple monocyte-macrophage subgroups, including classical monocytes (Mo_01_CD14), non-classical monocytes (Mono_like_03_CD14 CD16), M2-type macrophages (Macrophages_M2_quantiseq), and several neutrophil subgroups (Neu_04_TXNIP, Neu_03_ISG15, Neu_02_S100A12) all exhibited significant negative correlations (Fig 9C), highlighting the complex regulatory role of myeloid immune cells in AML prognosis.

TMB analysis revealed an unexpected result: the low-risk group showed higher TMB levels (Fig 9D). Mutation landscape analysis revealed significant genomic heterogeneity between high and low-risk groups: NPM1 mutation frequency in the high-risk group (6%) was significantly lower than in the low-risk group (12%); meanwhile, TTN, BCORL1, and RUNX1 showed much higher mutation frequencies in the low-risk group (15%, 12%, and 12% respectively) compared to the high-risk group (Fig 9E and F).

Mutation co-occurrence analysis revealed risk group-specific genomic characteristics: the high-risk group showed significant co-occurrence patterns between EDC4-VARS2 and FLT3-DNMT3A (Fig 9G). In contrast, the low-risk group

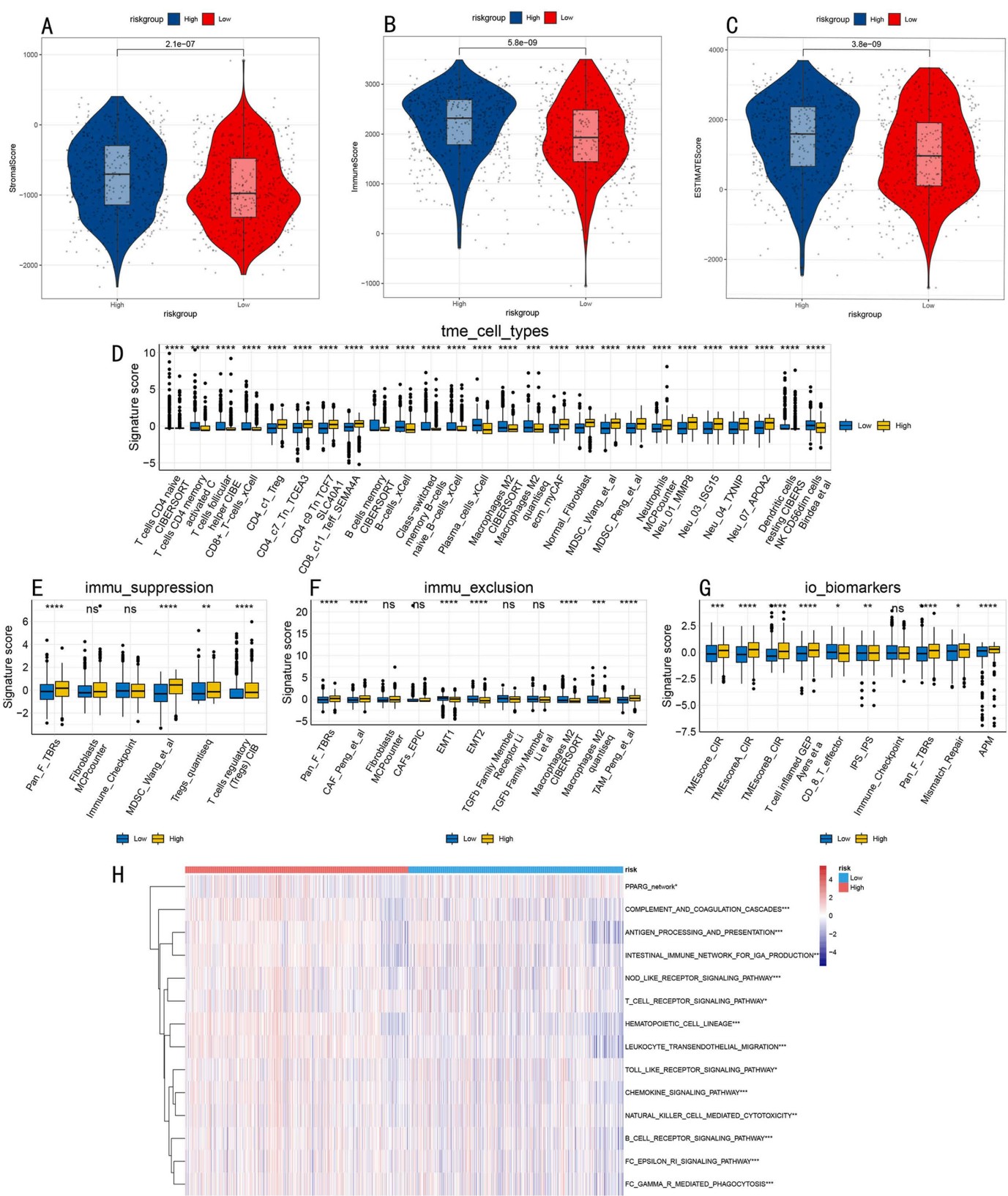

**Fig 7. Tumor Microenvironment Characteristic Analysis in META Integrated Dataset. (A-C).** Violin plots of microenvironment scores assessed by ESTIMATE algorithm, including stromal score **(A)**, immune score **(B)**, and comprehensive score **(C)**. **(D-G).** Box plots showing differences between high

and low-risk groups in immune cell types **(D)**, immune suppression **(E)**, immune rejection **(F)**, and biomarkers **(G)**. **(H)**. Hierarchical clustering heatmap of differential immune pathways, showing activity differences in 14 key pathways.

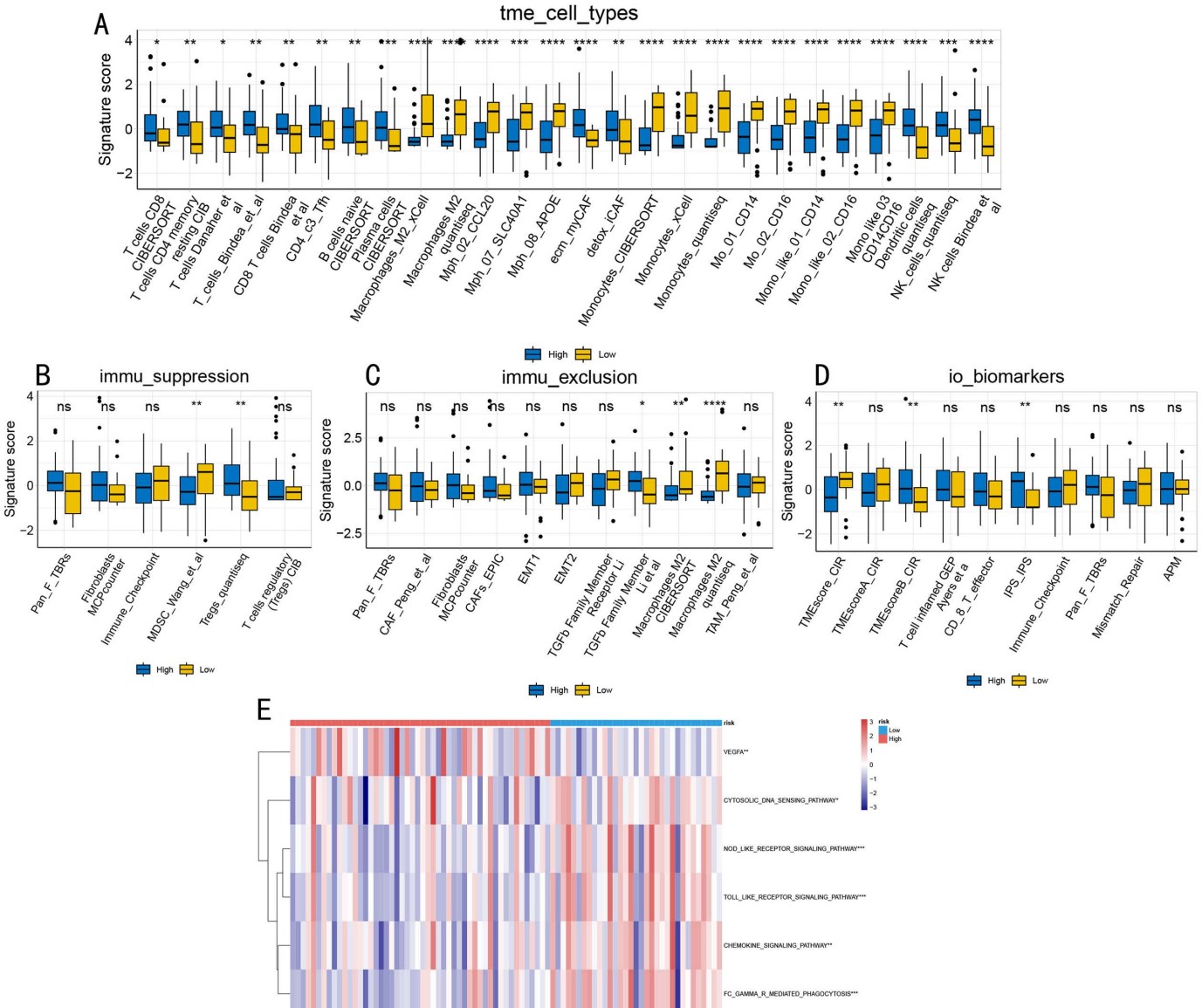

**Fig 8. Tumor Microenvironment Characteristic Analysis in TCGA-LAML Validation Set. (A)**. Differential analysis of immune cell types. **(B-D)**. Immune suppression **(B)**, immune rejection **(C)**, and biomarkers **(D)** in validation set. **(E)**. Hierarchical clustering heatmap of 6 key immune pathways in validation set. Statistical significance: *P < 0.05, **P < 0.01, ***P < 0.001, ****P < 0.0001.

exhibited a more complex mutation co-occurrence network, particularly significant co-occurrence relationships between LARP1, KMT2D, HIVEP3, and DNAH7 with multiple genes (including TTN, BCORL1, ACO1, etc.) (Fig 9H), suggesting that the low-risk group might possess a more complex but more stable genomic regulatory network.

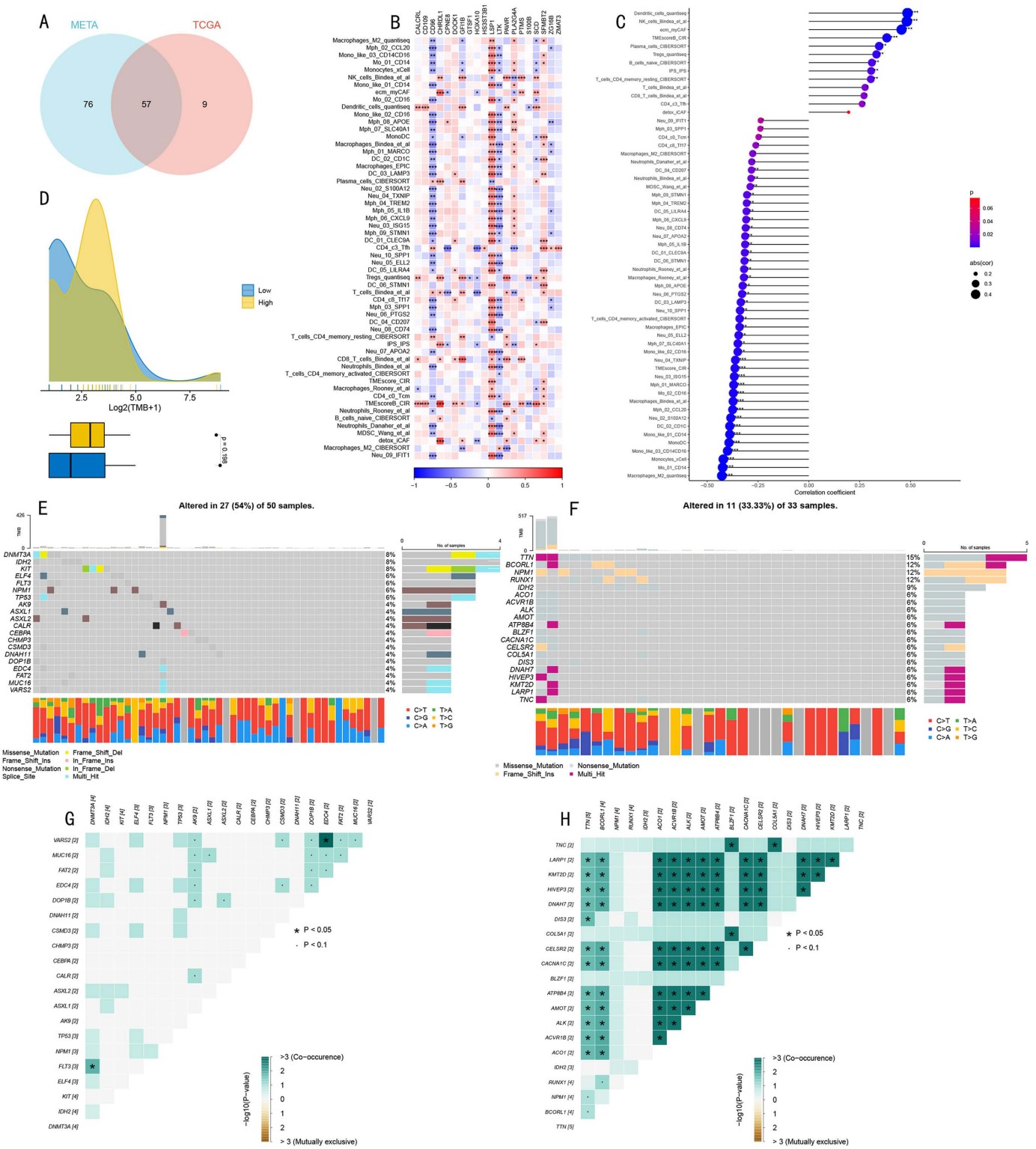

**Fig 9. Systematic Analysis of Somatic Mutation Spectrum and Tumor Heterogeneity. (A)**. Venn diagram of common features between META and TCGA datasets. **(B)**. Correlation heatmap between immune cell features and key predictive genes. **(C)**. Forest plot of quantitative analysis of feature gene correlations. **(D)**. Density plot and box plot comparison of TMB distribution. **(E, F)**. Mutation landscape waterfall plots for high-risk (E) and low-risk (F) groups. **(G, H)**. Mutation co-occurrence/mutual exclusivity relationship heatmaps for high-risk (G) and low-risk (H) groups.

## Drug sensitivity analysis and therapeutic implications

Based on molecular subtype differential characteristics, we first identified four significantly different signaling pathways through gene set enrichment analysis: Interferon Alpha Response、Myc Targets V1, Interferon Gamma Response, and Kras Signaling Up (Fig 10A). The differential activation of these pathways laid the molecular foundation for subsequent drug sensitivity analysis.

Through systematic IC50 value prediction, we identified eight potential therapeutic drugs showing significant sensitivity differences between high and low-risk groups. These drugs can be classified into four categories: epigenetic regulators: LAQ824 and MS-275 (histone deacetylase inhibitors, P = 0.000139 and P = 0.00104, respectively, Fig 10D and E); proteasome inhibitors: Bortezomib and MG-132 (P = 0.00747 and P = 0.0106, respectively, Fig 10B and H); kinase inhibitors: Imatinib (P = 0.000695, Fig 10C), Pazopanib (P = 0.000528, Fig 10G), and Tipifarnib (P = 0.0246, Fig 10I); and cytoskeleton-targeting drugs: Paclitaxel (P = 0.00015, Fig 10F). This providing important clues for personalized treatment strategies in patients with different risk stratification.

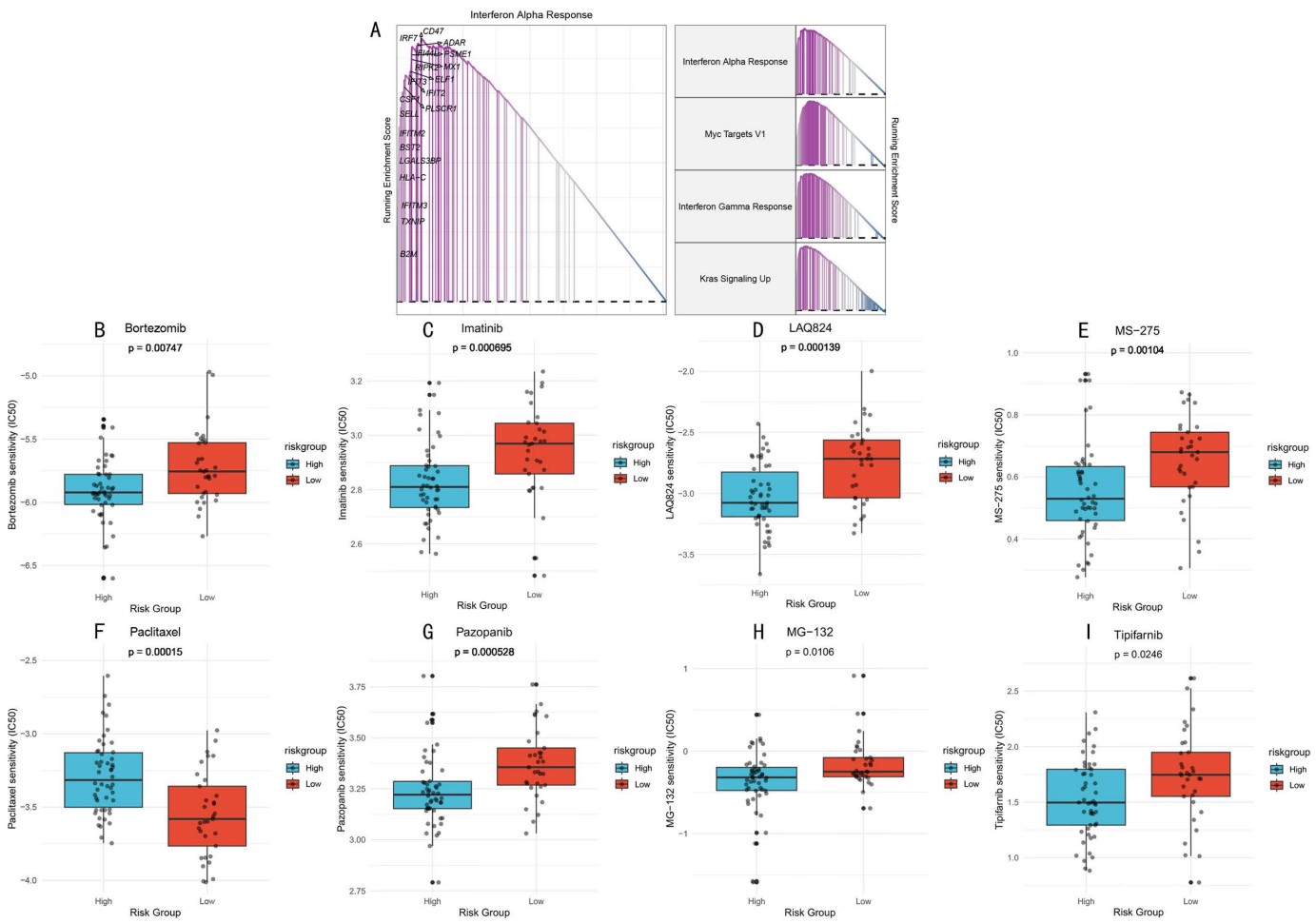

**Fig 10. Drug Sensitivity Analysis of High and Low-Risk Groups. (A)**. Gene enrichment analysis profiles of four differential pathways and their characteristic genes. (B-I). Sensitivity comparison of eight key therapeutic drugs between high and low-risk groups. Box plots show distribution of predicted IC50 values, with lower IC50 values indicating higher drug sensitivity. Statistical significance determined by Wilcoxon rank-sum test.

## Single-cell analysis in HSC of AML

To validate and further explore the biological significance of the RSF risk score model at the single-cell level, this study analyzed single-cell RNA sequencing data of AML patients obtained from the ABC portal website. UMAP visualization of the entire single-cell dataset revealed distinct clustering patterns of different cell types in the AML bone marrow micro-environment (Fig 11A). Cell type annotation identified multiple cell populations, including HSC, myeloid progenitor cells, monocytes, lymphocytes, and erythroid precursor cells, highlighting the complex cell composition in AML samples.

When cells were classified according to their malignant status (Fig 11B), malignant cells (red) and non-malignant cells (blue) showed differential distributions in the UMAP space. Malignant cells formed several distinct clusters, which may represent different leukemia clones or differentiation states. Given the crucial role of HSC in the pathogenesis of AML and its highest cell number and proportion, this study specifically extracted the HSC population for in-depth analysis. Notably, this study first used epigenetic regulation score calculation to reveal the UMAP visualization of the activity of epigenetic programs between malignant and non-malignant HSC (Fig 11C). It showed that malignant HSC (mainly in the red area) had significantly higher epigenetic scores compared to non-malignant HSC (mainly in the blue area), indicating that abnormal epigenetic regulation is a key feature of leukemia stem cells.

Statistical comparison confirmed that the epigenetic scores of malignant HSC were significantly higher than those of non-malignant HSC (P<0.0001, Fig 11D). This finding is consistent with the premise of this study and the overall RNA-seq observations, supporting the crucial role of epigenetic reprogramming in the pathogenesis of AML.

To validate the clinical relevance of the prognostic model of this study at the single-cell level, this study continued to calculate the LSC17 score and the RSF model score in HSC respectively. The LSC17 score, based on 17 leukemia stem cell-characteristic genes, was significantly higher in malignant HSC than in non-malignant HSC (P<0.0001, Fig 11E), confirming its ability to detect leukemia stem cell characteristics at the single-cell level. Most importantly, the RSF model score based on the top 20 prognostic genes also showed a significant distinction between malignant and non-malignant HSC (P<0.0001, Fig 11F). The significant increase in the RSF score in malignant HSC provided convincing evidence that the prognostic model of this study captured the fundamental biological characteristics of leukemia stem cells, which may explain its robust performance in predicting patient prognosis across multiple cohorts.

To further evaluate the performance of epigenetic scores, LSC17 scores, and RSF scores in distinguishing malignant from non-malignant HSC, this study divided cells into high- and low- expression groups based on the median of each score, and constructed contingency tables and performance index comparisons (Tables 2 and 3). Table 2 shows the association between the grouping based on different scoring methods and the malignant status of cells. All three scoring methods showed a higher proportion of high-score groups in malignant HSC, while low-score groups were dominant in non-malignant HSC. The RSF score identified 2,060 high-score cells in malignant cells (accounting for approximately 82.70% of the total number of malignant cells), and correctly classified 12,356 low-score cells in non-malignant cells (accounting for approximately 53.53% of the total number of non-malignant cells). Table 3 quantitatively compared the diagnostic performance of the three scoring methods. The results showed that the RSF score outperformed the epigenetic score and the LSC17 score in multiple indicators such as sensitivity (82.70%), specificity (53.53%), precision (16.11%), F1- score (26.98%), accuracy (56.59%), and negative predictive value (96.63%), indicating that the RSF score model not only has excellent prognostic prediction ability at the overall cohort level but also has high discriminative efficacy in distinguishing malignant from non-malignant HSC at the single-cell level. These findings further validate the ability of the RSF model to capture the biological characteristics of AML from the perspective of single-cell resolution, providing a more solid theoretical basis for its clinical application.

Each scoring method was divided into high- and low-score groups based on the median. The table shows the cross-distribution of different groups and the malignant status of cells (malignant vs non-malignant). The RSF score identified the largest number of high-score cells in malignant HSC (2,060) and correctly classified the largest number of low-score cells in non-malignant HSC (12,356).

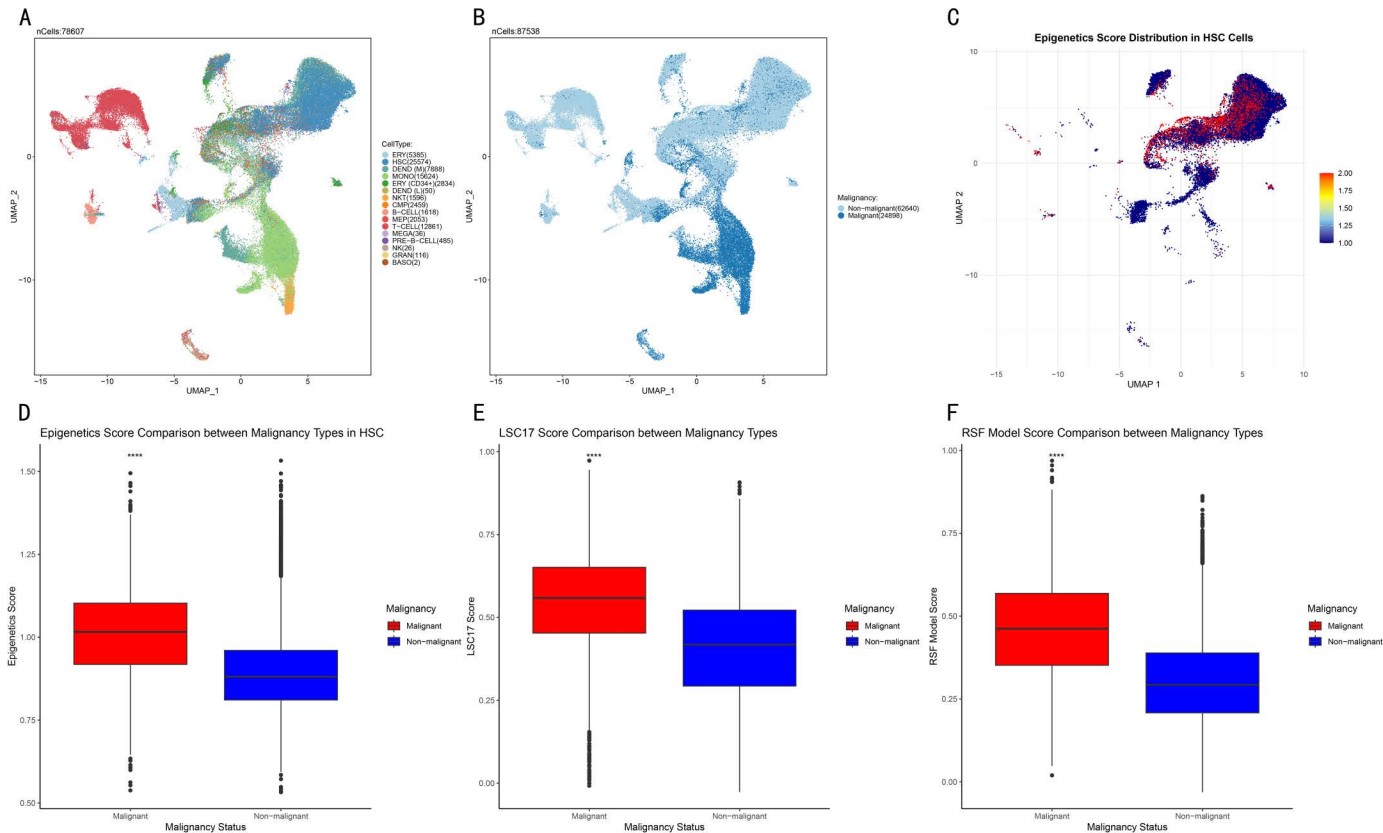

**Fig 11. Single-cell analysis reveals abnormal epigenetic regulation and activation of prognostic features in malignant hematopoietic stem cells. A.** UMAP visualization of all cells colored by cell type annotation, showing the heterogeneous cell composition in AML bone marrow. **B.** UMAP visualization colored by malignant status (red: malignant, blue: non-malignant), revealing different distribution patterns. **C.** UMAP visualization of the distribution of epigenetic scores in the HSC population, with color intensity representing the score level (red: high, blue: low). **D.** Box-plot comparison of epigenetic scores between malignant and non-malignant HSC, showing significantly higher scores in malignant cells (****P < 0.0001). **E.** Box-plot comparison of LSC17 scores between malignant and non-malignant HSC, demonstrating a significant increase in LSC17 features in malignant cells (****P < 0.0001). **F.** Box-plot comparison of the RSF model scores based on the top 20 prognostic genes between malignant and non-malignant HSC, showing significantly higher scores in malignant cells (****P < 0.0001).

**Table 2. Contingency table of score grouping and malignant status.**

| Scoring Method | Group | Malignant | Non-malignant | Total |
|---|---|---|---|---|
| Epigenetics | High | 2,001 | 10,786 | 12,787 |
| | Low | 490 | 12,297 | 12,787 |
| | Total | 2,491 | 23,083 | 25,574 |
| LSC17 | High | 1,958 | 10,829 | 12,787 |
| | Low | 533 | 12,254 | 12,787 |
| | Total | 2,491 | 23,083 | 25,574 |
| RSF | High | 2,060 | 10,727 | 12,787 |
| | Low | 431 | 12,356 | 12,787 |
| | Total | 2,491 | 23,083 | 25,574 |

**Table 3. Comparison of performance indicators of three scoring methods.**

| Scoring Method | Sensitivity | Specificity | Precision | F1 Score | Accuracy | Positive Predictive Value | Negative Predictive Value |
|---|---|---|---|---|---|---|---|
| Epigenetics | 80.32% | 53.26% | 15.64% | 26.27% | 56.16% | 15.64% | 96.17% |
| LSC17 | 78.63% | 53.09% | 15.31% | 25.63% | 55.63% | 15.31% | 95.84% |
| RSF | 82.70% | 53.53% | 16.11% | 26.98% | 56.59% | 16.11% | 96.63% |

This table shows multiple performance indicators of epigenetic scores, LSC17 scores, and RSF scores in distinguishing malignant from non-malignant HSC. The RSF score performed best (marked in green) in multiple indicators such as sensitivity (82.70%), specificity (53.53%), precision (16.11%), F1-score (26.98%), accuracy (56.59%), positive predictive value (16.11%), and negative predictive value (96.63%), demonstrating its superior discriminative ability at the single-cell level.

## Discussion

### Main findings and innovation

Through integrated analysis of multi-omics data from five independent cohorts, we established a novel epigenetic-based molecular classification system for AML, identifying two distinct subtypes (CS1 and CS2) with significant prognostic differences (P<0.001). The molecular basis of this classification is supported by systematic characterization of subtype-specific regulatory networks, particularly the dual-regulatory pattern involving core hematopoietic transcription factors (CEBPA, HOXA7/9, RUNX1) and epigenetic modifiers (BRD4, DOT1L, DNMT3A). This regulatory framework provides mechanistic insights into how epigenetic alterations influence leukemic cell fate through modulation of lineage-specific transcription programs. Our RSF model, incorporating 20 key predictive features including CPNE8, CD109, and CHRDL1, demonstrated superior prognostic accuracy compared to 82 alternative machine learning approaches, achieving consistent C-index values above 0.7 across multiple validation cohorts. This robust performance validates the clinical utility of our epigenetic-based classification system.

A particularly innovative aspect of our study is the comprehensive characterization of risk group-specific immune microenvironment features and their therapeutic implications. High-risk patients exhibited elevated immune and stromal scores (P<0.001) with an immunosuppressive phenotype dominated by M2 macrophages and Tregs, while low-risk patients showed enhanced NK cell activity and balanced immune cell compositions. These immune characteristics correlated with distinct mutation patterns, notably lower NPM1 mutation frequency (6% vs 12%) and specific co-mutation networks in high-risk groups. Furthermore, drug sensitivity analysis revealed subtype-specific therapeutic vulnerabilities: low-risk groups showed enhanced sensitivity to epigenetic regulators (LAQ824, MS-275; P<0.001) and proteasome inhibitors (Bortezomib, MG-132; P<0.01), while high-risk groups demonstrated unique sensitivity to Paclitaxel (P<0.001). These findings not only advance our understanding of AML heterogeneity but also provide a rational basis for personalized treatment strategies based on molecular subtype and risk stratification.

### Comparison with existing research

Compared to existing AML research, this study demonstrates unique advantages in classification systems, prognostic prediction, and epigenetic integration. The traditional WHO classification system primarily based on cell morphology, immunophenotype, and cytogenetic characteristics, though updated in 2022 to include some molecular markers, still fails to fully reflect disease epigenetic characteristics [40]. In contrast, our study, through integrating multi-level epigenetic information including DNA methylation, non-coding RNA, and histone modifications [41], not only expands classification dimensions but also reveals dynamic regulatory mechanisms of disease progression. This multi-omics integration-based typing method shows unique advantages in prognostic prediction, particularly demonstrating high specificity and sensitivity in identifying patients with favorable prognosis in the CS2 subtype.

Several AML prognostic models based on gene expression or mutation characteristics have been reported in recent years. The LSC17 score, developed by Stanley et al. [42], which measures leukemic stem cell activity through a 17-gene expression signature, and the genomics-based prognostic stratification system by Papaemmanuil et al. [43] have both been widely applied in clinical practice. Our comprehensive analysis revealed that while the LSC17 score showed significant prognostic value (P < 0.01), it demonstrated a lower hazard ratio compared to our RSF model in univariate Cox analysis. Importantly, through multivariate analysis, we found that both LSC17 score and our epigenetic-based risk score maintained independent prognostic value, suggesting they capture different aspects of disease biology. This led to our development of an integrated nomogram incorporating both scores, which showed superior predictive accuracy as evidenced by excellent calibration curves for 1-, 3-, and 5-year survival predictions. The dynamic analysis of time-dependent C-index demonstrated that this integrated approach consistently outperformed single prognostic factors throughout the follow-up period, addressing a key limitation of existing single-omics models.

Unlike previous models focused on single omics levels, our RSF model integrates epigenetic markers with traditional prognostic factors, enabling more comprehensive capture of disease heterogeneity. This integration is particularly valuable given our finding that epigenetic regulation patterns correlate strongly with immune microenvironment characteristics and drug sensitivity profiles. Notably, our model shows better predictive accuracy in distinctive high-risk patient populations, where traditional models often fall short. The improved performance in these challenging patient subgroups, combined with our ability to predict differential drug sensitivities, provides crucial information for clinical decision-making, especially in selecting between standard chemotherapy, targeted therapy, or clinical trial enrollment. This comprehensive prognostic approach represents a significant advance toward more personalized treatment strategies in AML.

Regarding epigenetic typing, another significant advantage of our study lies in adopting empirical Bayesian integrated clustering methods [44], which effectively handle the heterogeneity and complexity of multi-omics data. Compared to previous studies focusing only on single epigenetic markers (such as DNA methylation) [5], our approach more comprehensively characterizes molecular features of the disease by simultaneously considering synergistic effects of multiple epigenetic modifications. Furthermore, this study systematically analyzes the association between epigenetic typing and immune microenvironment for the first time, revealing the important role of epigenetic regulation in shaping the tumor immune microenvironment, a finding not fully reported in existing literature.

Moreover, compared with traditional whole-tissue-based analyses, the single-cell RNA sequencing analysis in this study provides an unprecedented perspective on cellular heterogeneity. Existing AML studies are mostly based on whole-tissue samples, which are unable to distinguish the contributions of malignant and non-malignant cells, and it is also difficult to capture the characteristics of key cell subsets. Although the single-cell studies by Wu et al. [45] and Ediriwickrema et al. [46] described the cellular heterogeneity of AML, they failed to integrate epigenetic regulation with prognostic prediction models. This study is the first to validate the biological basis of the RSF prognostic model at the single-cell level, demonstrating that it can effectively distinguish malignant from non-malignant HSC with a sensitivity of 82.70%, outperforming the established LSC17 scoring system. Notably, the single-cell analysis in this study revealed a significant increase in epigenetic scores in malignant HSC, providing direct evidence for the conclusion drawn from the multi-omics level in this study that "epigenetic regulation plays a central role in the pathogenesis of AML", which cannot be achieved by traditional whole-tissue analyses.

Through systematic comparison with existing research, this study not only validates the importance of known molecular markers but also identifies a series of new prognostic-related features, providing new insights for precision diagnosis and treatment of AML. Particularly in the construction and functional annotation of epigenetic regulatory networks, this study's depth and breadth of analysis surpass previous research, providing a more complete picture for understanding disease mechanisms. These findings not only complement existing classification systems but also provide important evidence for developing new therapeutic strategies.

### In-depth exploration of immune microenvironment

Regarding the immune microenvironment, this study found that the high-risk group has significantly elevated immune and stromal scores, yet this immune cell infiltration does not translate into effective anti-tumor responses. Detailed analysis revealed significantly increased proportions of M2-type macrophages and regulatory T cells (Tregs) in the high-risk group, accompanied by high expression of multiple immune checkpoint molecules (such as PD-L1, LAG3), suggesting the formation of an immunosuppressive microenvironment. This "cold" immune microenvironment may be one of the key factors leading to poor prognosis [47]. In contrast, the low-risk group shows more balanced immune cell components and more active anti-tumor immune responses, particularly maintaining relatively intact NK cell and cytotoxic T cell functions [48].

Mutation spectrum analysis revealed the genomic basis of risk stratification. The higher TMB level in the low-risk group appears to correlate with better prognosis, and this apparent contradiction may reflect a stronger neoantigen load and more effective immune surveillance. Particularly, the high frequency of NPM1 mutations in the low-risk group (12% vs 6%) suggests its protective role, consistent with previous studies [49]. The FLT3-DNMT3A co-mutation pattern observed in the high-risk group may promote disease progression through synergistic effects [50], providing new insights for developing targeted therapeutic strategies.

At the signaling pathway level, our study identified four distinctly activated pathways (Interferon Alpha Response, Myc Targets V1, Interferon Gamma Response, and Kras Signaling Up) that form an interconnected regulatory network defining molecular subtypes. Notably, the differential activation of interferon response pathways emerges as a central mechanism distinguishing the two subtypes, potentially explaining their distinct immune microenvironment characteristics [51]. The concurrent activation of Myc Targets V1 pathway, a key regulator of cell proliferation and metabolic reprogramming, suggests a complex interplay between immune response and cellular metabolism in determining disease progression [52]. Integrated analysis revealed that the activation states of these pathways are closely associated with specific epigenetic modification patterns, particularly evident in the differential sensitivity to epigenetic regulators (LAQ824 and MS-275) between risk groups. This observation suggests the existence of epigenetic-transcriptional regulatory feedback loops that maintain subtype-specific pathway activities [53]. Furthermore, we observed significant correlation between Kras Signaling Up pathway activation and specific immune cell infiltration patterns, particularly the enrichment of macrophages and Tregs in high-risk groups. This finding aligns with recent studies demonstrating the role of Kras signaling in immune evasion [54], providing new mechanistic insights into how oncogenic signaling pathways may contribute to microenvironment remodeling in AML. The identification of these pathway interactions not only deepens our understanding of AML pathogenesis but also suggests potential combination therapeutic strategies targeting both signaling pathways and epigenetic regulation.

These mechanistic findings not only deepen our understanding of AML heterogeneity but also provide a theoretical foundation for developing new therapeutic strategies. In particular, the close association between epigenetic regulation and immune microenvironment suggests that combining epigenetic regulators with immunotherapy might be a promising therapeutic strategy. Meanwhile, personalized treatment plans based on mutation spectrum and pathway activation characteristics may be key to improving therapeutic efficacy.

### Clinical translation, limitations and future directions

The prognostic stratification system developed in this study demonstrates significant value in clinical applications. The RSF model-based risk score not only accurately predicts patient prognosis but also provides objective evidence for treatment decisions. For patients classified as high-risk, more aggressive treatment strategies may need to be considered, such as early hematopoietic stem cell transplantation or enrollment in clinical trials. For low-risk patients, they may benefit from more moderate treatment regimens, thereby avoiding overtreatment. Notably, our drug sensitivity analysis results provide important references for individualized medication. For example, low-risk group patients show higher sensitivity to epigenetic regulators (such as LAQ824 and MS-275), which may guide clinical drug selection.

However, this study also has several limitations. First, despite integrating multiple cohorts, the sample size remains relatively limited, particularly in studying certain rare molecular subtypes. Second, the use of retrospective data may introduce selection bias and lacks prospective clinical validation. Technically, while multi-omics data integration provides a more comprehensive perspective, it also brings challenges in data processing and standardization. Furthermore, the lack of in vitro and in vivo functional experimental validation limits causal relationship determinations. The third limitation is the ethnic diversity of the study population. The datasets used in this study mainly originated from Western populations. The genetic and epigenetic backgrounds of different ethnic groups can vary, which may affect the applicability of our risk groups. Furthermore, the lack of in vitro and in vivo functional experimental validation limits causal relationship determinations.

Based on these findings and limitations, we propose several important future research directions: (1) Conducting large-scale, prospective, and international clinical validation studies to further confirm the clinical value of molecular typing and prognostic models; (2) Conducting in-depth functional studies, particularly exploring molecular mechanisms between epigenetic modifications and immune microenvironment remodeling; (3) Developing therapeutic strategies targeting newly discovered targets, especially considering combining epigenetic regulators with immunotherapy; (4) Optimizing prediction models, possibly through integrating more clinical features and dynamic monitoring data to improve prediction accuracy; (5) Validating drug sensitivity findings experimentally. For example, we should test drugs like LAQ824, MS-275, Bortezomib, and MG-132 on AML cell lines (in vitro) and mouse models (in vivo); (6) Establishing standardized multicenter research networks to promote standardized application of molecular typing in clinical practice. These studies will further advance the development of precision medicine in AML, ultimately improving patient outcomes.

## Conclusion

Through multi-omics integrated analysis, this study revealed two epigenetic molecular subtypes of AML with significant prognostic differences, established a robust prognostic prediction model, and elucidated the associative characteristics between epigenetic regulation and immune microenvironment. The research results provide new theoretical foundations for AML molecular typing while also providing important evidence for developing individualized treatment strategies. These findings not only deepen the understanding of AML molecular mechanisms but also provide new insights for precision medicine practice. Future research will focus on prospective validation of molecular typing, the development of novel therapeutic targets, and optimization of prediction models, aiming to further improve treatment outcomes for AML patients.

## Acknowledgments

We appreciated all the fellows in our department for their excellent work.

## Author contributions

**Conceptualization:** Shengyue Wang.

**Data curation:** Jincan Li, Shengyue Wang.

**Formal analysis:** Jincan Li, Shengyue Wang.

**Investigation:** Jincan Li.

**Methodology:** Jincan Li.

**Resources:** Shengyue Wang.

**Software:** Jincan Li.

**Supervision:** Shengyue Wang.

**Validation:** Jincan Li, Shengyue Wang.

**Writing – original draft:** Jincan Li, Shengyue Wang.

**Writing – review & editing:** Jincan Li, Shengyue Wang.

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
