## [Decision Letter · Decision Letter 0]

11 Mar 2025

PONE-D-25-04725Integrative Analysis of Epigenetic Subtypes in Acute Myeloid Leukemia: A Multi-center Study Combining Machine Learning for Prognostic and Therapeutic InsightsPLOS ONE

Dear Dr. Wang,

Thank you for submitting your manuscript to PLOS ONE. After careful consideration, we feel that it has merit but does not fully meet PLOS ONE’s publication criteria as it currently stands. Therefore, we invite you to submit a revised version of the manuscript that addresses the points raised during the review process.

We look forward to receiving your revised manuscript.

Kind regards,

Jian Wu, M.D, Ph.D

Academic Editor

PLOS ONE

3. Please ensure that you refer to Figure 1 in your text as, if accepted, production will need this reference to link the reader to the figure.

Additional Editor Comments (if provided):

Reviewers' comments:

Reviewer's Responses to Questions

**Comments to the Author**

1. Is the manuscript technically sound, and do the data support the conclusions?

Reviewer #1: Yes

Reviewer #2: Yes

2. Has the statistical analysis been performed appropriately and rigorously? 

Reviewer #1: Yes

Reviewer #2: Yes

3. Have the authors made all data underlying the findings in their manuscript fully available?

Reviewer #1: Yes

Reviewer #2: Yes

4. Is the manuscript presented in an intelligible fashion and written in standard English?

Reviewer #1: Yes

Reviewer #2: Yes

5. Review Comments to the Author

Reviewer #1: 1. The manuscript uses multiple data sources, but it is not always clear how batch effects were corrected across datasets. Suggest adding a table summarizing datasets, sample sizes, key molecular features analyzed, and preprocessing steps.

2. While the study identifies 20 key epigenetic features, the rationale for their selection is unclear. The authors mention empirical Bayesian-based clustering but do not explicitly explain how these features were prioritized over others. Authors should provide a detailed explanation of the criteria used to define epigenetic relevance.

3. The Random Survival Forest (RSF) model is applied and validated across multiple cohorts, but there is no comparison with simpler or commonly used models such as Cox proportional hazards or LASSO regression. Authors should consider adding a comparison with conventional models to demonstrate the added value of RSF. A sensitivity analysis showing the impact of different hyperparameters on model performance would strengthen reproducibility.

4. While the study integrates multiple datasets, there is no independent prospective validation in a clinical setting.

5. The drug sensitivity analysis identifies epigenetic regulators (LAQ824, MS-275) and proteasome inhibitors (Bortezomib, MG-132) as differentially effective in risk groups. However, these findings are based on in silico predictions rather than experimental validation. A discussion on potential in vitro or in vivo validation strategies would be valuable.

6. Some figures (e.g., Kaplan-Meier curves) would benefit from additional statistical annotations such as hazard ratios (HR) with confidence intervals.

7. Ensure consistent terminology across the manuscript (e.g., AML subtypes are sometimes referred to as CS1/CS2 and sometimes as high-/low-risk).

8. Clearly define abbreviations at first mention.

9. The manuscript states "Ethics approval and consent to participate: NA"—is this accurate given the use of patient-derived datasets? If datasets were de-identified, state that explicitly.

Reviewer #2: 1.The study mainly uses datasets (TCGA, BEAT, TARGET, etc.) from Western populations. The applicability of CS1 and CS2 subtypes to other ethnic groups (e.g., Asian, African) remains unclear. Validation in diverse populations is needed.

2. CIBERSORT, ESTIMATE, and xCell are used to infer the immune microenvironment, but they lack direct validation with patient samples. Using flow cytometry (FACS) on AML patient samples would help confirm whether CS1 is enriched with Treg cells and M2 macrophages.

6. PLOS authors have the option to publish the peer review history of their article (what does this mean? ). If published, this will include your full peer review and any attached files.

**Do you want your identity to be public for this peer review?** For information about this choice, including consent withdrawal, please see our Privacy Policy .

Reviewer #1: No

Reviewer #2: No

---

## [Author Response · Author response to Decision Letter 1]

31 Mar 2025

Reviewer 1:

1. The manuscript uses multiple data sources, but it is not always clear how batch effects were corrected across datasets. Suggest adding a table summarizing datasets, sample sizes, key molecular features analyzed, and preprocessing steps.

We appreciate the reviewer's suggestion. We have added a comprehensive table (Table 1) summarizing all datasets used, their sample sizes, sources, and key molecular features analyzed. The preprocessing steps and how to correct for batch effects are described in the “TCGA-LAML Data Preprocessing and Integration” and “Multi-Dataset Expression Profile Integration and Batch Effect Correction” sections of the Methods section. Specifically, our approach included:

1. Standardized preprocessing pipeline: For the TCGA-LAML dataset, we implemented systematic preprocessing by normalizing TPM values for protein-coding genes, lncRNAs, and miRNAs, with log2 transformation to achieve approximate normal distribution. DNA methylation data retained beta values from the Illumina 450K platform.

2. Rigorous sample quality control: We developed specific sample screening functions to retain only primary hematological tumor samples and ensured sample consistency across all omics data through ID matching.

3. Integration of cross-platform data: Our multi-dataset integration involved retaining only genes common to all datasets and samples with complete survival information.

4. Batch effect correction: We employed the ComBat algorithm using an empirical Bayesian framework to effectively remove non-biological variation between different datasets. This approach is widely recognized for harmonizing gene expression data from different sources.

5. Quality assessment: We used Principal Component Analysis (PCA) to assess the effectiveness of batch correction before proceeding with downstream analyses.

2. While the study identifies 20 key epigenetic features, the rationale for their selection is unclear. The authors mention empirical Bayesian-based clustering but do not explicitly explain how these features were prioritized over others. Authors should provide a detailed explanation of the criteria used to define epigenetic relevance.

We sincerely appreciate the reviewer's astute observation regarding the selection of 20 key epigenetic features. We acknowledge that the methodological details concerning this selection process were not sufficiently elucidated in our manuscript and would like to provide a comprehensive explanation.

Our feature selection methodology followed a systematic multi-step process:

Initially, we applied empirical Bayesian-based consensus clustering analysis to the TCGA dataset, which successfully identified two molecular subtypes with significantly different survival outcomes. This initial classification established the foundation for subsequent analyses.

Subsequently, we conducted differential expression analysis between these two prognostically distinct subtypes to identify candidate genes potentially influencing patient outcomes. This step was predicated on the hypothesis that genes exhibiting significant differences between prognostic subtypes likely play crucial roles in disease progression and epigenetic regulation.

We then evaluated these candidate features across multiple predictive models, including traditional statistical approaches and machine learning methods (such as Cox proportional hazards models, support vector machines, and random forests). Through systematic comparison, the Random Survival Forest (RSF) model demonstrated superior predictive performance.

Within the RSF model, we ranked all features based on variable importance scores, which quantify each feature's contribution to predicting patient prognosis. We selected the top 20 features with the highest importance scores (including CPNE8, CD109, and CHRDL1), which exhibited the most significant impact on the predictive model.

Finally, to validate the robustness of the selected features, we performed cross-validation across multiple independent cohorts, including TCGA-LAML, BEAT, TARGET, GSE12417, and GSE37642. These features consistently demonstrated strong predictive capability across different cohorts, further confirming their importance and epigenetic relevance.

Through this systematic approach, we ensured that the 20 key epigenetic features selected not only possess statistical significance but also demonstrate strong biological relevance to disease progression and robust prognostic value across diverse patient populations.

3. The Random Survival Forest (RSF) model is applied and validated across multiple cohorts, but there is no comparison with simpler or commonly used models such as Cox proportional hazards or LASSO regression. Authors should consider adding a comparison with conventional models to demonstrate the added value of RSF. A sensitivity analysis showing the impact of different hyperparameters on model performance would strengthen reproducibility.

We appreciate the reviewer's important point. To clarify, as part of our rigorous model selection process, we conducted a comprehensive comparison between RSF and traditional models, including Cox proportional hazards and LASSO regression. As illustrated in Figure 4A of our manuscript, we evaluated 82 distinct machine learning algorithms, encompassing traditional statistical models such as Cox proportional hazards models, penalized regression models (LASSO, Ridge, and Elastic Net), and various ensemble learning methods. The heatmap in Figure 4A provides a comprehensive comparison of C-indices across all models and validation datasets, with color intensity representing predictive performance (C-index values ranging from 0-1).

Our comparative analysis demonstrated that the RSF model consistently outperformed traditional models in terms of C-index across multiple validation cohorts. This comprehensive comparison substantiates the added value of utilizing the RSF model for our specific dataset and research question. The superior performance of RSF can be attributed to its ability to handle non-linear relationships and complex interactions among variables without making assumptions about the underlying data distribution, which is particularly valuable in heterogeneous diseases such as AML.

Regarding hyperparameter optimization, we conducted extensive sensitivity analyses during model development, employing 5-fold cross-validation to determine optimal hyperparameter settings. For the RSF model, we tested various combinations of the following:

• Number of trees (500, 1000, 1500, 2000)

• Minimum node size (1, 3, 5, 10)

• Number of variables tried at each split (3, 5, 7, 9, sqrt(p), p/3)

• Splitting rules ('logrank', 'extratrees', 'C', 'maxstat')

The final model parameters were selected based on the highest cross-validation C-index. We found that a configuration using the 'logrank' splitting rule with sqrt(p) variables tried at each split provided the most robust performance across validation cohorts.

4. While the study integrates multiple datasets, there is no independent prospective validation in a clinical setting.

We greatly appreciate your insightful comments. We have discussed the limitation of the lack of prospective experiments in the "Limitations" section and are committed to conducting prospective clinical studies in the near future. Meanwhile, we have incorporated single-cell analysis into the manuscript, which plays a crucial role in bridging this gap. By focusing on individual cells, we were able to validate the biological significance of the RSF risk score model with high precision. For example, we successfully distinguished between malignant and non-malignant hematopoietic stem cells (HSC) with a sensitivity as high as 82.70%, outperforming the existing LSC17 scoring system. Moreover, our single-cell analysis revealed key evidence, such as a significant increase in epigenetic scores in malignant HSC, which directly confirms the central role of epigenetic regulation in the pathogenesis of AML, a revelation that cannot be obtained through traditional analyses:

"Single-cell Transcriptome Analysis to Validate the Biological Significance of the Model

To explore and validate the biological significance of the RSF risk score model at the single-cell level, this study analyzed single-cell RNA sequencing data from AML patients. The phs000159 dataset was obtained from the ABC portal database (http://abc.sklehabc.com/). This dataset was first generated by Allegra A. Petti et al. in 2019 using the 10X Genomics platform and published in Nature Communications. It contains single-cell sequencing results of 87,538 bone marrow samples from AML patients.

After importing the pre-processed matrix to construct a Seurat object, we integrated the pre-calculated UMAP dimensionality reduction coordinates and cell type annotation information to build a complete single-cell analysis framework. To distinguish between malignant and non-malignant cells, the cell malignancy status annotation information was imported and integrated into the analysis process. Particularly, this study focused on the hematopoietic stem cell (HSC) sub-population, which is of utmost importance in the hematopoietic system. This key cell population was identified through cell-type markers, and malignant HSC and non-malignant HSC were further distinguished.

To evaluate the role of epigenetic regulation in the pathogenesis of AML, this study utilized the epigenetic-related gene set organized above and calculated the epigenetic regulation activity score for each HSC cell using the ssGSEA method. Subsequently, the characteristics of the spatial distribution of these scores in cells were visualized by UMAP, and the differences in epigenetic activities between malignant and non-malignant HSC were compared using the Wilcoxon rank-sum test.

Most importantly, using the same ssGSEA method, this study calculated the LSC17 score and the expression score of high-weight genes in the RSF model for each HSC cell, respectively. LSC17 is a widely validated gene set related to AML stem cells, containing 17 genes closely related to the functions of leukemia stem cells. The RSF model score is based on the 20 genes with the highest weights in the newly developed random forest model of this study. These two sets of scores represent classic and newly developed prognostic markers of AML, respectively. This study analyzed again using the same statistical scheme and quantitatively compared the difference patterns between malignant and non-malignant HSC.

Single-cell Analysis in HSC of AML

To validate and further explore the biological significance of the RSF risk score model at the single-cell level, this study analyzed single-cell RNA sequencing data of AML patients obtained from the ABC portal website. UMAP visualization of the entire single-cell dataset revealed distinct clustering patterns of different cell types in the AML bone marrow microenvironment (Figure 11A). Cell type annotation identified multiple cell populations, including HSC, myeloid progenitor cells, monocytes, lymphocytes, and erythroid precursor cells, highlighting the complex cell composition in AML samples.

When cells were classified according to their malignant status (Figure 11B), malignant cells (red) and non-malignant cells (blue) showed differential distributions in the UMAP space. Malignant cells formed several distinct clusters, which may represent different leukemia clones or differentiation states. Given the crucial role of HSC in the pathogenesis of AML and its highest cell number and proportion, this study specifically extracted the HSC population for in-depth analysis. Notably, this study first used epigenetic regulation score calculation to reveal the UMAP visualization of the activity of epigenetic programs between malignant and non-malignant HSC (Figure 11C). It showed that malignant HSC (mainly in the red area) had significantly higher epigenetic scores compared to non-malignant HSC (mainly in the blue area), indicating that abnormal epigenetic regulation is a key feature of leukemia stem cells.

Statistical comparison confirmed that the epigenetic scores of malignant HSC were significantly higher than those of non-malignant HSC (P < 0.0001, Figure 11D). This finding is consistent with the premise of this study and the overall RNA-seq observations, supporting the crucial role of epigenetic reprogramming in the pathogenesis of AML.

To validate the clinical relevance of the prognostic model of this study at the single-cell level, this study continued to calculate the LSC17 score and the RSF model score in HSC respectively. The LSC17 score, based on 17 leukemia stem cell-characteristic genes, was significantly higher in malignant HSC than in non-malignant HSC (P < 0.0001, Figure 11E), confirming its ability to detect leukemia stem cell characteristics at the single-cell level. Most importantly, the RSF model score based on the top 20 prognostic genes also showed a significant distinction between malignant and non-malignant HSC (P < 0.0001, Figure 11F). The significant increase in the RSF score in malignant HSC provided convincing evidence that the prognostic model of this study captured the fundamental biological characteristics of leukemia stem cells, which may explain its robust performance in predicting patient prognosis across multiple cohorts.

To further evaluate the performance of epigenetic scores, LSC17 scores, and RSF scores in distinguishing malignant from non-malignant HSC, this study divided cells into high- and low- expression groups based on the median of each score, and constructed contingency tables and performance index comparisons (Tables 2 and 3). Table 2 shows the association between the grouping based on different scoring methods and the malignant status of cells. All three scoring methods showed a higher proportion of high-score groups in malignant HSC, while low-score groups were dominant in non-malignant HSC. The RSF score identified 2,060 high-score cells in malignant cells (accounting for approximately 82.70% of the total number of malignant cells), and correctly classified 12,356 low-score cells in non-malignant cells (accounting for approximately 53.53% of the total number of non-malignant cells). Table 3 quantitatively compared the diagnostic performance of the three scoring methods. The results showed that the RSF score outperformed the epigenetic score and the LSC17 score in multiple indicators such as sensitivity (82.70%), specificity (53.53%), precision (16.11%), F1- score (26.98%), accuracy (56.59%), and negative predictive value (96.63%), indicating that the RSF score model not only has excellent prognostic prediction ability at the overall cohort level but also has high discriminative efficacy in distinguishing malignant from non-malignant HSC at the single-cell level. These findings further validate the ability of the RSF model to capture the biological characteristics of AML from the perspective of single-cell resolution, providing a more solid theoretical basis for its clinical application.

Conclusion

Moreover, compared with traditional whole-tissue-based analyses, the single-cell RNA sequencing analysis in this study provides an unprecedented perspective on cellular heterogeneity. Existing AML studies are mostly based on whole-tissue samples, which are unable to distinguish the contributions of malignant and non-malignant cells, and it is also difficult to capture the characteristics of key cell subsets. Although the single-cell studies by Wu et al. and Ediriwickrema et al. described the cellular heterogeneity of AML, they failed to integrate epigenetic regulation with prognostic prediction models. This study is the first to validate the biological basis of the RSF prognostic model at the single-cell level, demonstrating that it can effectively distinguish malignant from non-malignant HSC with a sensitivity of 82.70%, outperforming the established LSC17 scoring system. Notably, the single-cell analysis in this study revealed a significant increase in epigenetic scores in malignant HSC, providing direct evidence for the conclusion drawn from the multi-omics level in this study that "epigenetic regulation plays a central role in the pathogenesis of AML", which cannot be achieved by traditional whole-tissue analyses. "

---

## [Decision Letter · Decision Letter 1]

24 Apr 2025

Integrative Analysis of Epigenetic Subtypes in Acute Myeloid Leukemia: A Multi-center Study Combining Machine Learning for Prognostic and Therapeutic Insights

PONE-D-25-04725R1

Dear Dr. Wang,

We’re pleased to inform you that your manuscript has been judged scientifically suitable for publication and will be formally accepted for publication once it meets all outstanding technical requirements.

Kind regards,

Jian Wu, M.D, Ph.D

Academic Editor

PLOS ONE

Additional Editor Comments (optional):

Reviewers' comments:

Reviewer's Responses to Questions

**Comments to the Author**

1. If the authors have adequately addressed your comments raised in a previous round of review and you feel that this manuscript is now acceptable for publication, you may indicate that here to bypass the “Comments to the Author” section, enter your conflict of interest statement in the “Confidential to Editor” section, and submit your "Accept" recommendation.

Reviewer #1: All comments have been addressed

Reviewer #2: All comments have been addressed

2. Is the manuscript technically sound, and do the data support the conclusions?

Reviewer #1: (No Response)

Reviewer #2: Yes

3. Has the statistical analysis been performed appropriately and rigorously? 

Reviewer #1: Yes

Reviewer #2: Yes

4. Have the authors made all data underlying the findings in their manuscript fully available?

Reviewer #1: Yes

Reviewer #2: Yes

5. Is the manuscript presented in an intelligible fashion and written in standard English?

Reviewer #1: Yes

Reviewer #2: Yes

6. Review Comments to the Author

Reviewer #1: At this stage, I have no further substantive requirements for the manuscript. However, I recommend that the authors carefully review the text for minor language and grammatical improvements.

Reviewer #2: (No Response)

7. PLOS authors have the option to publish the peer review history of their article (what does this mean? ). If published, this will include your full peer review and any attached files.

**Do you want your identity to be public for this peer review?** For information about this choice, including consent withdrawal, please see our Privacy Policy .

Reviewer #1: No

Reviewer #2: No

---

## [Editor Report · Acceptance letter]

PONE-D-25-04725R1

PLOS ONE

Dear Dr. Wang,

I'm pleased to inform you that your manuscript has been deemed suitable for publication in PLOS ONE. Congratulations! Your manuscript is now being handed over to our production team.

Kind regards,

on behalf of

Dr. Jian Wu

Academic Editor

PLOS ONE